# LEARNING EQUIVARIANT TENSOR FUNCTION REPRESENTATIONS VIA COVARIANT ALGEBRA OF BINARY FORMS

## ABSTRACT

Representing tensor-valued functions of tensor arguments is fundamental in many modeling problems. Tensor functions play a central role in constructing reduced-order approximations and are particularly useful for nonlinear anisotropic constitutive modeling of physical phenomena, such as fluid turbulence and material deformation among others. By imposing equivariance under the orthogonal group, tensor functions can be finitely and minimally generated by using the isomorphism between binary forms and symmetric trace-free tensors. After determining minimal generators, their coefficients can be learned as functions of the invariants of the tensor arguments by training on data which facilitates generality of the models. The algebraic nature of the learned models makes them interpretable by revealing underlying dynamics, and it keeps the models economical as they contain the theoretically minimum required number of terms. Determining minimal representations of higher-order tensor functions has remained computationally intractable in many cases of interest until now. The current work overcomes this limitation. Numerically efficient algorithms for generating tensor functions and reducing them to minimal sets are presented. A few classical tensor function representations and an approach to a bottleneck in modeling turbulence are worked out to showcase the practical applicability of our framework.

## 1 INTRODUCTION

Tensor function representations are important for building models of tensors using tensors arguments. They are useful for many applications in physics, particularly fluid turbulence (Alfonsi (2009); Speziale et al. (1991); Speziale (1990)) and continuum mechanics (Olive et al. (2017); Boehler & Boehler (1987); Boehler (1979)), and also modeling problems dealing with image and video data as tensors of order 2 and 3 (Vasilescu & Terzopoulos (2002); Shashua & Hazan (2005)). It is a much more intricate modeling problem than while working with scalars due to the complexity of the structure imposed by tensor symmetries.

For learning physically meaningful tensor function representations, in addition to maintaining consistent symmetry in index permutations, it is also important to preserve proper rotational or orthogonal equivariance, i.e., rotating or reflecting the inputs should also rotate or reflect the outputs predictably (Zheng (1994)).

$$\mathbf{T} : \mathbb{V} \to \mathbb{T}^m, \quad \text{where} \quad \mathbb{V} := \mathbb{T}^{n_1} \oplus \mathbb{T}^{n_2} \oplus \mathbb{T}^{n_3} \cdots \oplus \mathbb{T}^{n_k} \tag{1}$$

In this work, a new strategy is presented for building equivariant tensor functions of tensor of the type shown in Eq. 1. These functions can be modeled as a linear combination of Invariant scalar *coefficients* ($\mathbf{C}$s) and equivariant tensor monomials ($\mathbf{B}$s) which depend on some tensor arguments ($\mathbf{A}s$) as shown in Eq. 2. The $\mathbf{C}s$ are functions of joint invariants ($\lambda_i$) of $\mathbf{A}$s and the tensor monomials are *einsum* type tensor products of $\mathbf{A}$s.

$$\mathbf{T}(\mathbf{A}_1, \mathbf{A}_2, \cdots, \mathbf{A}_k) = \sum_{i=1}^{\gamma} C_i(\lambda_1, \lambda_2, \cdots, \lambda_m) \mathbf{B}_i(\mathbf{A}_1, \mathbf{A}_2, \cdots, \mathbf{A}_k) \tag{2}$$

The model in Eq. 2 is *complete* if 1) all invariants of $\mathbb{V}$ are *generated* by (functions of) $\lambda$s and 2) all tensor monomials in $\mathbb{V}$ are linear combinations of $\mathbf{B}$s under scalar coefficient functions of the joint invariants. Additionally, the model is *minimal* if removing any tensor monomial or joint invariant makes the model incomplete. Working with minimal generator sets significantly reduces the number of terms involved in modeling. In many cases it is not just preferable, but necessary to make computations tractable. The goal of the current work is to develop a framework for determining minimally complete tensor function representations which are equivariant under the $\mathrm{SO}(3, \mathbb{R})$ group, and to establish their applicability in modeling any generic physical tensor in 3 dimensions. From a modeling perspective, the structure (or the coded versions) of $\mathbf{B}_i$'s and the $\lambda_i$'s can be derived *apriori*, while the exact coefficients $C_i$s are *learned* using domain-specific data from $\mathbf{T}$ and $(\mathbf{A}_1, \mathbf{A}_2, \cdots, \mathbf{A}_k)$. Eq. 2 shows a general model with $\gamma$ terms. Algebraically, the joint invariants form an algebra (say, **Inv**), and for completeness of the model $\lambda_1, \lambda_2, \ldots, \lambda_m$ should be algebra generators for **Inv**. Meanwhile, equivariant tensor monomials form a submodule (say, **Cov**) of tensors (in which $\mathbf{T}$ exists) over **Inv**, and $(\mathbf{B}_1, \mathbf{B}_2, \ldots, \mathbf{B}_\gamma)$ should be module generators for **Cov**.

One of the earliest works in this vein is by Robertson (1940), which was widely used for developing isotropic tensor functions for closure modeling in turbulence. Motivated by some use cases in linear elasticity theory, other authors derived several minimally complete isotropic tensor function representations in Wang (1970); Smith (1971); Zheng (1993a; 1994; 1993b). However, these results were restricted to tensors of order less than or equal to 2. After significant dormancy, recently Olive & Auffray (2014) gave a minimal set of isotropic invariants of a symmetric third order tensor. These results are scalar function representations, but an extension to general tensor function representations was not made, until now. There has been considerable interest in learning such tensor function representations using Equivariant Neural Networks (ENNs) Villar et al. (2021); Gasteiger et al. (2020); Geiger & Smidt (2022) which have been used successfully in a range of applications (for example, Jumper et al. (2021); Satorras et al. (2021); Gregory et al. (2024)). We develop a new methodology for deriving algebraically minimal equivariant functions with more interpretability than black box ENNs architectures, which could be useful for similar applications.

The workflow for the current work is summarized in Fig. 3. A central idea is to work with binary forms for deriving the tensor monomials and the invariants, then learn the coefficient functions using neural networks. This is possible because: 1) every tensor can be irreducibly decomposed into a direct sum of symmetric trace free (i.e., harmonic) tensors (Zou et al. (2001); Spencer (1970)); and 2) There is an isomorphism (Smith & Bao (1997); Boehler et al. (1994)) between harmonic tensors and binary forms, by which tensor monomials can be derived in the space of binary forms. Another advantage of working with binary forms is that the equivariant polynomials in binary forms naturally form an *algebra* (and not a module), which can be finitely generated (Gordan (1868); Hilbert (1993)). Our results thus arise from the invariant theory of binary forms, which is a very mature field (Hilbert (1993); Olver (1999); Kung & Rota (1984)) and reliable algorithms (Sturmfels (2008); Bruns et al. (2017); Eisenbud et al. (2001)) for effective computations. Working in the binary forms space is not only convenient, but we found it essential for making computations tractable.

The rest of the paper is laid out as follows. Some background for invariant theory is introduced in §2. This will be used in §3 to discuss the overall methodology for deriving minimal tensor monomials and invariants. A supervised learning problem is discussed with tensor invariants as inputs and the monomial coefficients as outputs, which will complete the learning framework for general tensor function representations. In §4, the approach is verified by deriving symmetric tensor function representations of one and two symmetric second order tensors, for which we compare against classical results of Wang (1970). In §5, the fourth order Rapid Pressure Strain Rate (RPSR) correlation tensor (Pope (2001)), a well-known bottleneck (Kassinos et al. (2001)) in modeling turbulence, is explored using the current methodology. We present numerical results showing a substantial improvement over existing modeling approaches.

Contributions of the current work:

1. A new paradigm for learning equivariant tensor functions from tensor tuple inputs to tensor outputs is developed by first deriving the underlying algebraic structure, and next learning scalar coefficient functions using neural networks and problem specific data. This paradigm also enforces tensor trace constraints exactly the the final models.
2. Minimally complete tensor function representations are derived using covariant algebra of binary forms in an interpretable learning based setting.

3. Efficient numerical algorithms are developed for generating covariant modules and finding minimal module generators.
4. *Nonlinear* RSPR tensor models are built using turbulence structure tensors (Kassinos (1995)) for the first time and tested on a wide variety of turbulent flow configurations.

## 2 MATHEMATICAL BACKGROUND AND PREREQUISITES

Groups describe symmetries Olver (1999). We recall that a *group* $G$ is a set with a binary operation that is associative, has an identity element, and admits inverses. $GL(n, \mathbb{F}) = \{M \in \mathbb{F}^{n \times n}\}$ is the general linear group. Main examples in this work are the matrix groups $SO(3, \mathbb{R}) = \{R \in \mathbb{R}^{3 \times 3} : RR^\top = RR^\top = I\}$ and $SL(2, \mathbb{C}) = \{M \in \mathbb{C}^{2 \times 2} : \det(M) = 1\}$, where the group operation is matrix multiplication.

Representations encode how symmetries operate on spaces. A **W**-*representation* of $G$ is a finite-dimensional vector space (real or complex) on which $G$ acts by linear maps. That is, there exists a group homomorphism $\rho : G \to GL(\mathbf{W})$ from $G$ to invertible linear transformations of $\mathbf{W}$. Often we abbreviate $\rho(g)w$ as $g \cdot w$ for $g \in G$ and $w \in \mathbf{W}$ with $\rho$ being tacitly understood. Key examples of representations are spaces of tensors, 3-dimensional in each direction, under the action of 3D rotations. For instance, the space $\mathbf{W}$ of $3 \times 3$ real matrices is a representation of $SO(3, \mathbb{R})$ where the action is by conjugation: $R \cdot W := RWR^\top$ for $R \in \mathrm{SO}(3, \mathbb{R})$ and $W \in \mathbf{W}$.

**Definition 2.1** (*G*-equivariant and *G*-invariant functions[1]). Let $\mathbf{W}$ and $\mathbf{V}$ be representations of some group $G$. A function $B : \mathbf{W} \to \mathbf{V}$ is said to be *equivariant* with respect to $G$ if:

$$B(g \cdot W) = g \cdot B(W), \quad \forall g \in G, \quad \forall W \in \mathbf{W}. \tag{3}$$

Further, let $\mathbf{X}$ be any set. A function $\lambda : \mathbf{W} \to \mathbf{X}$ is said to be *invariant* with respect to $G$ if:

$$\lambda(g \cdot W) = \lambda(W), \quad \forall g \in G, \quad \forall W \in \mathbf{W}. \tag{4}$$

**Definition 2.2** (*G*-covariant module and *G*-invariant algebra). Let $\mathbf{W}$ and $\mathbf{V}$ be $K$-representations of some group $G$, where $K = \mathbb{R}$ or $K = \mathbb{C}$. Then the *covariant module* is $\mathbf{Cov}_G(\mathbf{W}, \mathbf{V})$ is the set of all $G$-equivariant polynomial functions over $K$ from $\mathbf{W}$ to $\mathbf{V}$. Further, the *invariant algebra* $\mathbf{Inv}_G(\mathbf{W})$ is the set of all $G$-invariant polynomial functions over $K$ from $\mathbf{W}$ to $K$.

We note that $\mathbf{Inv}_G(\mathbf{W})$ is closed under multiplication and addition, while $\mathbf{Cov}_G(\mathbf{W}, \mathbf{V})$ is closed under addition and scalar multiplication by elements of $\mathbf{Inv}_G(\mathbf{W})$. This makes $\mathbf{Inv}_G(\mathbf{W})$ into an algebra, and $\mathbf{Cov}_G(\mathbf{W}, \mathbf{V})$ into a module over this algebra. We are interested in this paper chiefly in $\mathrm{SO}(3, \mathbb{R})$-equivariant and -invariant functions between tensor spaces.

A helpful relationship exists between the groups $SO(3, \mathbb{R})$ and $SL(2, \mathbb{C})$. Namely, there is a double cover (see Appendix D for explicit details):

$$\Psi : SL(2, \mathbb{C}) \to SO(3, \mathbb{C}), \tag{5}$$

where $SO(3, \mathbb{C})$ is the complexification of $SO(3, \mathbb{R})$. Double cover here means that $\Psi$ is a surjective homomorphism that is 2-to-1. This relationship lets us identify real representations of $SO(3, \mathbb{R})$ with complex representations of $SL(2, \mathbb{C})$, via $\mathbf{W} \mapsto \mathbf{W} \otimes_{\mathbb{R}} \mathbb{C}$.

**Definition 2.3.** Let $S_n$ denote the complex vector space of *binary forms* of order $n$. For $\mathbf{f} \in S_n$, it can be written

$$\mathbf{f}(x, y) = \sum_{i=0}^{n} \binom{n}{i} a_i x^{n-i} y^i, \quad \text{for } (x, y) \in \mathbb{C}^2. \tag{6}$$

A *space of binary forms*, $\mathbf{V}$, is defined as follows (where $\oplus$ denotes direct sum):

$$\mathbf{V} = \bigoplus_{i=0}^{s} S_{n_i}, \quad n_i \in \mathbb{N}. \tag{7}$$

Spaces of binary forms are representations of $SL(2, \mathbb{C})$ under change of variables. A binary form *covariant* is an $SL(2, \mathbb{C})$-equivariant function between two binary form spaces. It can be written as

---

[1]The terms functions and maps are used interchangeably.

a polynomial in $x, y$ and the coefficients of binary forms. Similarly, a binary form *invariant* is an $SL(2, \mathbb{C})$-invariant polynomial function. Gordan (1868) and Hilbert (1993) proved that the *algebra* of covariants for a binary form space, $\mathbf{Cov}(\mathbf{V}) := \oplus_{n=0}^{\infty} \mathbf{Cov}_{SL(2,\mathbb{C})}(\mathbf{V}, S_n)$, is finitely generated.

There is a natural *bi-grading* in covariant algebra, i.e., each covariant can be graded using two integers: the *order* and the *degree*. The order refers to the sum of powers of $\mathbf{x}, \mathbf{y}$ and the degree refers to the sum of powers of coefficients of any term in the binary forms. For a bihomogeneous binary form, the degree and order on all terms will be the same, hence can be used for uniquely grading them. For example, $\mathbf{f} \in S_4$ is an order 4, degree 1 covariant, while $\mathbf{f}^2$ is an order 8, degree 2. The order 0 covariants coincide with the invariants (Hilbert (1993)), hence the invariant algebra is contained in the covariant algebra.

It is known that the space of $S_n$ is an $SL(2, \mathbb{C})$ *irreducible* representation, which cannot be further decomposed into direct sum of representations. The $SL(2, \mathbb{C})$ irreducible decomposition of a tensor product is given by *Clebsh-Gordan decomposition* (Olive (2017)),

$$S_n \otimes S_p \cong \bigoplus_{r=0}^{\min(n,p)} S_{n+p-2r}, \tag{8}$$

where $\bigoplus$ indicates a direct sum over the binary form spaces. For each $0 \le r \le \min(n, p)$, up to nonzero scalar, there is a unique *transvectant*. Let $\mathbf{f} \in S_m, \mathbf{g} \in S_n$, their transvectant of index $r$, is denoted by $(\mathbf{f}, \mathbf{g})_r$, and given as follows where $\pi_r$ denotes $SL(2, \mathbb{C})$-equivariant projection:

$$\pi_r : S_m \otimes S_n \to S_{m+n-2r}, \quad \mathbf{f} \otimes \mathbf{g} \mapsto (\mathbf{f}, \mathbf{g})_r := \pi_r(\mathbf{f} \otimes \mathbf{g}). \tag{9}$$

**Lemma 2.1.** *Olive & Auffray (2014) The transvectant operation of index $r$ between two binary forms $\mathbf{f} \in S_m$ and $\mathbf{g} \in S_n$ is given by*

$$(\mathbf{f}, \mathbf{g})_r = \frac{(m-n)!}{m!} \frac{(n-r)!}{n!} \sum_{i=0}^{r} (-1)^i \binom{r}{i} \frac{\partial^r \mathbf{f}}{\partial^{r-i} x \partial^i y} \frac{\partial^r \mathbf{g}}{\partial^i x \partial^{r-i} y}. \tag{10}$$

**Remark.** *Let $d_f, o_f$ be the degree and orders of $\mathbf{f}$, then for $(\mathbf{f}, \mathbf{g})_r$, the degree is $d_f + d_g$ and its order is $o_f + o_g - 2r$.*

## 3 TENSOR REPRESENTATION METHODS

Let $\mathbb{T}^n = (\mathbb{R}^3)^{\otimes n}$ and let $\mathbf{T} \in \mathbb{T}^n$ be a general order $n$ tensor. Equation 2 shows a general tensor function representation of $\mathbf{T}$ using $(\mathbf{A}_1, \mathbf{A}_2, \cdots, \mathbf{A}_k)$ where $\mathbf{A}_i \in \mathbb{T}^{o(i)}$,

$$\mathbf{T}(\mathbf{A}_1, \mathbf{A}_2, \cdots, \mathbf{A}_k) = \sum_{i=1}^{\gamma} C_i(\lambda_1, \lambda_2, \cdots, \lambda_m) \mathbf{B}_i(\mathbf{A}_1, \mathbf{A}_2, \cdots, \mathbf{A}_k).$$

For completeness of this model, $\lambda_i$'s must generate the invariant algebra $\mathbf{Inv}_{SO(3,\mathbb{R})}(\mathbb{T}^{o(1)} \oplus \mathbb{T}^{o(2)} \oplus \cdots \oplus \mathbb{T}^{o(k)})$ and $\mathbf{B_i}'s$ must generate $\mathbf{Cov}_{SO(3)}(\mathbb{T}^{o(1)} \oplus \mathbb{T}^{o(2)} \oplus \cdots \oplus \mathbb{T}^{o(k)}, \mathbb{T}^n)$ as an module over the invariant ring. From definition 2.2, it follows that if $\lambda_i$'s and the $\mathbf{B}_i$'s are minimal generators, then the model $\mathbf{T}(\mathbf{A}_1, \mathbf{A}_2, \cdots, \mathbf{A}_k)$ is minimally complete. Once the model is developed, the coefficients functions $C_i$'s must be learned as functions(Robertson (1940)) of the invariant ring generators ($\lambda_i$'s) using neural networks and domain-specific training data. Here, we first show the link between general tensors and harmonic tensors. Then we show how to accomplish the three major steps involved in deriving tensor function representations: 1) Using Harmonic Tensors (§3.1) to derive the minimal invariant algebra generators (§3.2); 2) Obtaining the minimal covariant module generators (§3.2); and 3) learning coefficient functions using neural networks (§3.4). Additionally, §3.3 shows an algorithm to reduce to *minimal* module generators.

### 3.1 IRREDUCIBLE DECOMPOSITION INTO HARMONIC TENSORS

The space of harmonic (i.e., symmetric and trace free) tensors $\mathbb{H}^n \subset \mathbb{T}^n$ is an $SO(3, \mathbb{R})$ irreducible representation. For the space of general order $n$ tensors, $\mathbb{T}^n$, the following theorem holds:

**Theorem 3.1.** $\mathbb{T}^n$ *irreducibly decomposes as* $\mathbb{H}^n \oplus \mathbb{H}^{n-1} + \cdots \oplus \mathbb{H}^0$. *(Spencer (1970); Zou et al. (2001))*

**Corollary 3.1.1.** *Covariant spaces between general tensor spaces can be expressed in terms of covariant spaces between harmonic tensor spaces.*

*Proof.* From Theorem 3.1,

$$\mathbf{Cov}_G \left( \bigoplus_{i=1}^k \mathbb{T}^{o(i)}, \mathbb{T}^n \right) \cong \mathbf{Cov}_G(\mathbb{T}^{o(1)} \oplus \mathbb{T}^{o(2)} \oplus \cdots \oplus \mathbb{T}^{o(k)}, \mathbb{T}^n)$$

$$\cong \mathbf{Cov}_G \left( \bigoplus_{l=0}^{o(1)} \mathbb{H}^l \oplus \bigoplus_{l=0}^{o(2)} \mathbb{H}^l \oplus \cdots \oplus \bigoplus_{l=0}^{o(k)} \mathbb{H}^l, \bigoplus_{i=0}^n \mathbb{H}^i \right)$$

$$\cong \mathbf{Cov}_G \left( \bigoplus_{j=1}^k \bigoplus_{l=0}^{o(j)} \mathbb{H}^l, \bigoplus_{i=0}^n \mathbb{H}^i \right) \cong \bigoplus_{i=0}^n \mathbf{Cov}_G \left( \bigoplus_{j=1}^k \bigoplus_{l=0}^{o(j)} \mathbb{H}^l, \mathbb{H}^i \right).$$

This completes the proof. $\qquad\qquad\square$

For example, the symmetric order 2 tensor space $\mathbb{S}\mathrm{ym}^2 \subset \mathbb{T}^2$ can be decomposed into:

$$\mathbb{S}\mathrm{ym}^2 \simeq \mathbb{H}^2 \oplus \mathbb{H}^0$$

$$\mathbf{Cov}_{SL(2,\mathbb{C})}(\mathbb{S}\mathrm{ym}^2, \mathbb{S}\mathrm{ym}^2) \simeq \mathbf{Cov}_{SL(2,\mathbb{C})}(\mathbb{H}^2 \oplus \mathbb{H}^0, \mathbb{H}^2 \oplus \mathbb{H}^0)$$

$$\mathbf{Cov}_{SL(2,\mathbb{C})}(\mathbb{S}\mathrm{ym}^2, \mathbb{S}\mathrm{ym}^2) \simeq \mathbf{Cov}_{SL(2,\mathbb{C})}(\mathbb{H}^2 \oplus \mathbb{H}^0, \mathbb{H}^2) \oplus \mathbf{Cov}_{SL(2,\mathbb{C})}(\mathbb{H}^2 \oplus \mathbb{H}^0, \mathbb{H}^0). \quad (11)$$

To develop complete tensor function representations for $\mathbf{Cov}_{SO(3,\mathbb{R})}(\mathbb{S}\mathrm{ym}^2, \mathbb{S}\mathrm{ym}^2)$, the invariant generators of $\mathbb{H}^2 \oplus \mathbb{H}^0$ and the $\mathbf{Cov}_{SO(3,\mathbb{R})}(\mathbb{H}^2 \oplus \mathbb{H}^0, \mathbb{H}^2)$ and $\mathbf{Cov}_{SO(3,\mathbb{R})}(\mathbb{H}^2 \oplus \mathbb{H}^0, \mathbb{H}^0)$ are required. This example is revisited and worked out in §4.1.Corollary 3.1.1 is a useful result which implies that **Inv** and **Cov** for harmonic tensors are sufficient to build tensor function representations for any general tensor. Below we show that a finite set of generators can be derived.

## 3.2 HARMONIC TENSORS AND BINARY FORMS

There is an association (Zheng (1994); Boehler et al. (1994)) from real irreducible representations of $SO(3,\mathbb{R})$ to complex irreducible representations of $SL(2,\mathbb{C})$. Via equation 5, there is an isomorphism as $SL(2,\mathbb{C})$ representations:

$$\mathbb{H}^n \otimes \mathbb{C} \cong S_{2n}. \quad (12)$$

This link is well known in the field of constitutive modeling and used to derive the isotropic invariants of traceless symmetric order 3 and order 4 tensors (Smith & Bao (1997)) and more recently a symmetric order 3 tensor Olive & Auffray (2014). It was also used to derive the minimal integrity basis for the order 4 elasticity tensor (Olive et al. (2017)). We use this link to show two important results.

**Theorem 3.2.** *Complete $SO(3,\mathbb{R})$ equivariant tensor function representations can be derived from $SL(2,\mathbb{C})$ covariant modules[2] of binary forms.*

*Proof.* Using Eq. 12,

$$\mathbf{Inv}_{SO(3,\mathbb{R})} \left( \bigoplus_{i=1}^k \mathbb{H}^{n_i} \right) \otimes \mathbb{C} \cong \mathbf{Inv}_{SL(2,\mathbb{C})} \left( \bigoplus_{i=1}^k S_{2n_i} \right) \quad (13)$$

In addition to these two invariant rings are isomorphic, we can say that for any order $m$, the following covariant modules are isomorphic:

$$\mathbf{Cov}_{SO(3,\mathbb{R})} \left( \bigoplus_{i=1}^k \mathbb{H}^{n_i}, \mathbb{H}^m \right) \otimes \mathbb{C} \cong \mathbf{Cov}_{SL(2,\mathbb{C})} \left( \bigoplus_{i=1}^k S_{2n_i}, S_{2m} \right) \quad (14)$$

$\qquad\qquad\square$

---

[2]The direct finite sum of covariant modules gives the covariant algebra.

**Remark.** *This concludes theoretical development in the current work and brings about an important comment. Using Theorem 3.2 and Corollary 3.1.1, the tensor function representations in Eq. 2 can be built, provided the invariant and covariant generators of binary forms are known:*

$$\mathbf{Cov}_{\mathrm{SO}(3,\mathbb{R})}\left(\bigoplus_{i=1}^{k}\mathbb{T}^{o(i)},\mathbb{T}^n\right)\cong\bigoplus_{i=0}^{n}\mathbf{Cov}_{\mathrm{SL}(2,\mathbb{C})}\left(\bigoplus_{j=1}^{k}\bigoplus_{l=0}^{o(j)}S_{2l},S_{2i}\right). \tag{15}$$

### 3.3 EFFECTIVE NUMERICAL COMPUTATIONS

The generators for $\mathbf{Inv}_G$ and $\mathbf{Cov}_G$ of binary forms can be finitely obtained using the Gordan's Algorithm (Gordan (1868); Olive (2017) which is explained in detail in Appendix E. For any space of binary forms $\mathbf{V} := \bigoplus_{i=1}^{k} S_{2i}$, the algorithm gives finite generators for any $S_n$ covariant module, $\mathbf{Cov}_{\mathrm{SL}(2,\mathbb{C})}(\mathbf{V}, S_n)$. It can also be used to generate the invariant algebra as $\mathbf{Inv}_G(\mathbf{V}) \simeq \mathbf{Cov}_G(\mathbf{V}, S_0)$. Results for covariant algebra of $S_6 \oplus S_2, S_4 \oplus S_4, S_6 \oplus S_4 \oplus S_2$ and several others are computed using Gordan's algorithm and readily available in literature. We note that the generators computed by Gordan's algorithm are in the form of iterated transvectants as defined in equation 10, which facilitates their efficient evaluation. The results can be used to develop complete $SO(3,\mathbb{R})$ equivariant tensor functions. But these representations will not be minimal, because Gordan's algorithm does not provide minimal generators, only complete ones.

---

**Algorithm 1:** Identifying the minimal generating set for an R-module

**Inputs:** Generators of R-Module M, Generators of Ring R
**Result:** Minimal generating set of R-Module

1 **begin**
2      Initialize $p \leftarrow 65521$          `// Some large prime`
3      Initialize $K \leftarrow ZZ/p$          `// Galois (Finite) Field`
4      Initialize $R \leftarrow K[c_1, c_2, \ldots, c_k, x, y]$      `// Poly Ring with Coefs and Vars`
5      Initialize $\mathbf{I} \leftarrow \{i_1, i_2, \ldots, i_m\}$          `// Ring Generators`
6      Initialize $\mathbf{F} \leftarrow \{f_1, f_2, \ldots, f_n\}$          `// Module Generators`
7      $\mathbf{I}^+ \leftarrow \{i_l | \mathrm{degree}(i_l) > 0\}$
8      $\mathfrak{m} \leftarrow \mathrm{ideal}(\mathbf{I}^+)$          `// The Irrelevant (maximal) Ideal`
9      **for** $\forall f_l \in \mathbf{F}$ **do**
10          $f_l^{\mathfrak{m}} \leftarrow f_l \% \mathfrak{m}$          `// Viewing $M$ gens as $M/\mathfrak{m}M$ gens`
11      **end**
12      $\mathbf{F}^{\mathfrak{m}} \leftarrow \{f_1^{\mathfrak{m}}, f_2^{\mathfrak{m}}, \ldots, f_n^{\mathfrak{m}}\}$
13      $\gamma \leftarrow$ Linearly Independent Row Indices ($\mathbf{F}^{\mathfrak{m}}$)    `// Using Gaussian Elimination`
14      **return** $\mathbf{F}^{(min)} \leftarrow \{f_{\gamma(1)}, f_{\gamma(2)}, \ldots, f_{\gamma(s)}\}$          `// Minimal Gen. Set`
15 **end**

---

In order to remedy this, the *minimal* generator subsets of Gordan's generators for **Inv** and **Cov** must be identified. For reducing generators of **Inv** to a minimal set, Olive (2017) gives a degree-per-degree approach using the Hilbert series (Bedratyuk (2010)). As far as the authors are aware, there are fewer readily implemented methods for identifying minimal generators of a module over a ring (like **Cov** over **Inv**) from a finite generating set. So, the Algorithm 1 is proposed. This uses the graded version of Nakayama's Lemma ( Eisenbud (2013)) which expresses module generation in terms of vector space generation over a field, so that linear algebra can be used for minimal generator identification. The lemma says that if $\mathfrak{m}$ is the homogeneous maximal ideal of the graded ring $R$, then $M/\mathfrak{m}M$ is a vector space over the field $R/\mathfrak{m}$ and $\{f_1, f_2, \ldots\}$ form a module generating set for $M$ if and only if the reductions $\{f_1^{\mathfrak{m}}, f_2^{\mathfrak{m}}, \ldots\}$ form a vector space spanning set for $M/\mathfrak{m}M$ over $R/\mathfrak{m}$. Further, since the polynomials in our computations have rational coefficients, the linear algebra calculations can be done over a finite field by working modulo a prime number. We choose a few large primes and repeat the computations to ensure that the results are accurate.

The algorithm is as follows. Load the generators of the invariant ring and the covariant module upto some fixed degree using Gordan's algorithm. Create the maximal ideal $I^+$ ($\mathfrak{m}$) using positively graded ring elements. View the module generators, **f**, of M in $M/\mathfrak{m}M$ by taking the remainder

against $\mathfrak{m}$, denoted by $\mathbf{f}\%\mathfrak{m}$. Now, implement reduced row echelon form and identify the linearly independent entries which correspond to $R$-linearly independent $\mathbf{f}$'s; these are the minimal generators. If the invariant ring is not too large, the $\mathbf{f}\%\mathfrak{m}$ step could be computed using a Gröbner basis (Sturmfels (2008)). If that is too restrictive, the numerical linear algebra algorithm proposed in Appendix F can be used. These computations were done using Macaulay2 (Eisenbud et al. (2001)).

### 3.4 LEARNING THE COEFFICIENTS

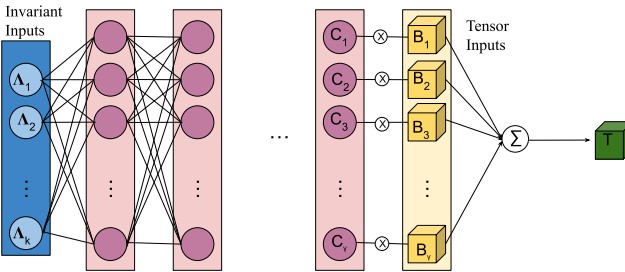

Figure 1: Schematic of the architecture for learning tensor representations as shown in Eq. 2. Circles are scalars ($\mathbb{H}^0$) and cubes are tensors ($\mathbb{H}^n$). $\mathbf{\Lambda}$'s are the invariants, $\mathbf{C}_i$'s are scalar functions of invariants, $\mathbf{B}_i$'s are the tensor monomials, $\mathbf{T}$ is the tensor function representation.

The minimal generators of $\mathbf{Inv_G}$ and $\mathbf{Cov_G}$ complete the learning architecture. The minimal generators will all be iterated polynomial transvectants, which must be converted back into $\mathbb{H}^n$ using the mapping shown in Appendix B. A schematic of the proposed architecture is shown in Fig. 3. The model has two input layers, the invariant algebra generator, labelled $\Lambda_i$ and the covariant module generators labeled $\mathbf{B}_i$ in yellow. The scalar coefficient functions are learned using a hidden layers, with appropriate size, based on the complexity of the problem. We can visualize the contributions from each each tensor input towards the modeling opening new possibilites for discovering insights about the modeling process. The covariant module computations can be reused for every run after the first time. An example is worked out in §5.

## 4 COMPUTATIONS AND RESULTS

### 4.1 MODELING A SYMMETRIC ORDER 2 TENSOR IN TERMS OF A SYMMETRIC ORDER 2 TENSOR

To model $\mathbf{A}(\mathbf{B})$, where, $\mathbf{A}, \mathbf{B} \in \mathbb{S}\text{ym}^2$, following Eq. 11, this problem is split into two sub parts. The covariant algebra of $S_4$ is well known(Olive (2017)). It is summarized in Table. 2 First minimal generators of $\mathbf{Inv}(S_4 \oplus S_0)$ are computed minimally. They are $(\mathbf{v}, \mathbf{v})_4$ and $(\mathbf{v}, (\mathbf{v}, \mathbf{v})_2)_4$, which correspond to $\text{tr}(\mathbf{B}^2)$ and $\text{tr}(\mathbf{B}^3)$ from $S_4$, and $S_0$ corresponds to $\text{tr}(B)$. These 3 invariants are inputs to coefficients and also used to create an R-module. Minimal generators of $\mathbf{Cov}(S_4 \oplus S_0, S_4)$ are $\mathbf{v}$ and the transvectant $(\mathbf{v}, \mathbf{v}_2)$, which correspond to $\mathbf{B}$ and $\mathbf{B}^2$. Putting all these together,

$$\mathbf{A}(\mathbf{B}) \in \mathbf{Cov}_{\text{SO}(3,\mathbb{R})}(\mathbb{H}^2 \oplus \mathbb{H}^0, \mathbb{H}^2 \oplus \mathbb{H}^0) \simeq \mathbf{Cov}_{\text{SL}(2,\mathbb{C})}(S_4 \oplus S_0, S_4 \oplus S_0)$$
$$\simeq \mathbf{Cov}_{\text{SL}(2,\mathbb{C})}(S_4 \oplus S_0, S_4) \oplus \mathbf{Cov}_{\text{SL}(2,\mathbb{C})}(S_4 \oplus S_0, S_0)$$

$$\mathbf{A}(\mathbf{B}) = \left( c_1 \mathbf{B} + c_2 \mathbf{B}^2 \right) + c_3 \mathbf{I}.$$

where $c_i = c_i(\text{tr}(\mathbf{B}), \text{tr}(\mathbf{B}^2), \text{tr}(\mathbf{B}^3))$ are to be learned using data from $\mathbf{A}$ and $\mathbf{B}$, in most cases using neural networks. These results can be verified from the results of Smith (1971); Zheng (1993a).

### 4.2 MODELING A SYMMETRIC ORDER 2 TENSOR USING TWO SYMMETRIC ORDER 2 TENSOR

The previous result is extended in this example. Let, $\mathbf{A}, \mathbf{B}, \mathbf{C} \in \mathbb{S}\text{ym}^2$, then $\mathbf{A}(\mathbf{B}, \mathbf{C})$ is derived. Irreducible decomposition shows,

| Deg | Covs | Harmonic Tensors |
|---|---|---|
| 1 | $\mathbf{q}, \mathbf{p}$ | $\mathbf{H}_{1a} := \mathbf{B}, \mathbf{H}_{1b} := \mathbf{C}$ |
| 2 | $(\mathbf{p}, \mathbf{q})_2$ | $\mathbf{H}_{2a} := (\mathbf{BC} + \mathbf{CA}) - \frac{1}{3}\text{tr}(\mathbf{BC} + \mathbf{CB})\mathbf{I}$ |
| | $(\mathbf{q}, \mathbf{q})_2, (\mathbf{p}, \mathbf{p})_2$ | $\mathbf{H}_{2b} := \mathbf{B}^2 - \frac{1}{3}\text{tr}(\mathbf{B^2})\mathbf{I}, \mathbf{H}_{2c} := \mathbf{C}^2 - \frac{1}{3}\text{tr}(\mathbf{C^2})\mathbf{I}$ |
| 3 | $(\mathbf{q}, (\mathbf{p}, \mathbf{p})_2)_2$ | $\mathbf{H}_{3a} := (\mathbf{BC}^2 + \mathbf{C}^2\mathbf{B}) - \frac{2}{3}\text{tr}(\mathbf{C}^2)\mathbf{H}_{1a} - \frac{1}{3}\text{tr}(\mathbf{H}_{2c}\mathbf{H}_{1a} + \mathbf{H}_{1a}\mathbf{H}_{2c})\mathbf{I}$ |
| | $(\mathbf{p}, (\mathbf{q}, \mathbf{q})_2)_2$ | $\mathbf{H}_{3b} := (\mathbf{CB}^2 + \mathbf{B}^2\mathbf{C}) - \frac{2}{3}\text{tr}(\mathbf{B}^2)\mathbf{H}_{1b} - \frac{1}{3}\text{tr}(\mathbf{H}_{2b}\mathbf{H}_{1b} + \mathbf{H}_{1b}\mathbf{H}_{2b})\mathbf{I}$ |
| 4 | $[(\mathbf{p}, \mathbf{q})_3]^2$ | $\mathbf{H}_4 := 6(\mathbf{B}^2\mathbf{C}^2 + \mathbf{C}^2\mathbf{B}^2) + 10\mathbf{H}_{1a}^2\text{tr}(\mathbf{C}^2) +$ $10\mathbf{H}_{1b}^2\text{tr}(\mathbf{B}^2) + 2\text{tr}(\mathbf{C}^2)\text{tr}(\mathbf{B}^2)\mathbf{I}$ $-\frac{1}{3}(6\text{tr}(\mathbf{C}^2\mathbf{B}^2) + 6\text{tr}(\mathbf{B}^2\mathbf{C}^2) + 26\text{tr}(\mathbf{B}^2)\text{tr}(\mathbf{C}^2))\mathbf{I}$ |

Table 1: Minimal module generators of order $4$ covariant polynomials expressed as iterated transvectants and their corresponding expressions as harmonic tensors in $\mathbb{H}^2$.

$$\mathbf{A}(\mathbf{B}, \mathbf{C}) \in \mathbf{Cov}_{SO(3,\mathbb{R})}(2\mathbb{H}^2 \oplus 2\mathbb{H}^0, \mathbb{H}^2 \oplus \mathbb{H}^0)$$
$$\simeq \mathbf{Cov}_{SL(2,\mathbb{C})}(2S_4, S_4) \oplus \mathbf{Cov}_{SL(2,\mathbb{C})}(2S_4, S_0) \oplus \mathbf{Cov}_{SL(2,\mathbb{C})}(2S_0, S_4) \oplus \mathbf{Cov}_{SL(2,\mathbb{C})}(2S_0, S_0)$$

$\mathbf{Cov}_{SL(2,\mathbb{C})}(2S_0, S_4)$ will have no terms. The terms from $\mathbf{Cov}_{SL(2,\mathbb{C})}(2S_4, S_0)$, and $\mathbf{Cov}_{SL(2,\mathbb{C})}(2S_0, S_0)$ will be reduced to $c\mathbf{I}$ in the model, for some coefficient function $c$. To build the tensor function representations for $\mathbf{Cov}_{SL(2,\mathbb{C})}(2S_4, S_4)$, the order $4$ module generators are build using the non-invariant covariant algebra of $S_4 \oplus S_4$, whose covariant algebra is derived in Olive (2017) is summarized in Table. 3. By solving a linear Diophantine system for every degree, 43 order 4 module generators were identified. Using Algorithm 1, they are reduced to a linearly independent minimal set of 8 polynomials. The linearly independent covariant polynomials and the corresponding harmonic tensors are written in Table 1. Using these generators, a model for a symmetric tensor can be written as,

$$\mathbf{A}(\mathbf{B}, \mathbf{C}) = c_1\mathbf{I} + c_2\mathbf{B} + c_3\mathbf{C} + c_4(\mathbf{BC} + \mathbf{CB}) + c_5\mathbf{B}^2 + c_6\mathbf{C}^2$$
$$+c_7(\mathbf{BC}^2 + \mathbf{C}^2\mathbf{B}) + c_8(\mathbf{CB}^2 + \mathbf{B}^2\mathbf{C}) + c_9(\mathbf{C}^2\mathbf{B}^2 + \mathbf{B}^2\mathbf{C}^2) \quad (16)$$

These 9 terms are consistent with the terms given by Wang (1970) and also noted in Smith (1970).

## 5 RESULTS FOR TURBULENCE MODELING

### 5.1 MODELING THE RAPID PRESSURE STRAIN RATE CORRELATION

Direct numerical simulations of fluid turbulence is prohibitively expensive in complex geometries. In practice, it is common to simulate by approximately modeling the effects of turbulence. One particularly difficult aspect to model is the Rapid Pressure Strain Rate (RSPR, $\mathbf{\Pi}$) tensor plays a central role in re-distrubuting the turbulence stresses. It has been extensively studied (Launder et al. (1975); Johansson & Hallbäck (1994); Girimaji (2000)) and it is a well-known bottleneck in turbulence modeling. The RSPR tensor is defined in Eq. 17 using the mean velocity gradient $U_{i,j} := \frac{\partial U_i}{\partial x_j}$ and an order $\mathbf{M} \in \mathbb{T}^4$ tensor.

$$\Pi_{ij} = 2U_{n,m}(M_{imnj} + M_{jmni}) \quad (17)$$

The mean velocity gradients are generally known apriori and the modeling is only focused on the $\mathbf{M}$ tensor. It is hypothesized that the $\mathbf{M}$ tensor depends on three turbulence structure tensorsKassinos et al. (2001): Reynolds Stress ($\mathbf{R} \in \mathbb{S}\text{ym}^2$), Dimensionality ($\mathbf{D} \in \mathbb{S}\text{ym}^2$), Stropholysis ($\mathbf{Q}^h \in \mathbb{H}^3$). Physically, $\mathbf{R}$ quantifies the anisotropy in the velocity components, $\mathbf{D}$ quantified the anisotropy in

the turbulent length scales and $\mathbf{Q}^h$ quantifies how strongly flow rotation breaks reflectional symmetry. Consequently, a model for $\mathbf{M}(\mathbf{R}, \mathbf{D}, \mathbf{Q}^h)$ is valuable for turbulence modeling. This is first reformulated as a problem of modeling $\mathbf{M}^h(\mathbf{R}^h, \mathbf{D}^h, \mathbf{Q}^h)$ where, $\mathbf{R}^h$ and $\mathbf{D}^h$ are the harmonic projections of $\mathbf{R}$ and $\mathbf{D}$, which are commonly known as Reynolds Stress anisotropy ($\mathbf{R}^h \in \mathbb{H}^2$) and Dimensionality anisostopy ($\mathbf{D}^h \in \mathbb{H}^2$) in the field turbulence modelingChoi & Lumley (2001). First, the exact decomposition of $\mathbf{M}$ carried out by KassinosKassinos (1995) is noted:

$$M_{ijkl} = M_{(ijkl)} + \frac{1}{8}\left[4M_{ijkl} - 4M_{klij}\right] + \frac{1}{12}[4M_{lkij} - 4M_{kjil} - 4M_{ilkj}$$
$$-M_{ljki} - M_{iklj} - M_{kilj}] \tag{18}$$

$\mathbf{M}$ must satisfy some constraints on its traces: $M_{iikl} = D_{kl}, M_{klii} = R_{kl}, \epsilon_{its}M_{sptj} = Q_{ipj}$. These are introduced in Eq. 18 which simplifies to Eq. 19. By modeling the $\mathbf{M}^h$ and substituting in the harmonic decomposition (Eq. 19), trace constraints are satisfied *exactly* for the model of $\mathbf{M}$.

$$M_{ijkl} = \left[M_{ijkl}^h + \frac{1}{7}\left(\delta_{(ij}R_{kl)} + \delta_{(ij}D_{kl)}\right) - \frac{1}{35}\delta_{(ij}\delta_{kl)}R_{ii}\right] + \frac{1}{2}\left[\epsilon_{zkj}Q_{zil}^h - \epsilon_{zil}Q_{zkj}^h\right] +$$
$$\frac{1}{6}\left[(\delta_{il}\delta_{jk} + \delta_{ik}\delta_{lj} - 2\delta_{ij}\delta_{kl})R_{ii} + 3(\delta_{kl}R_{ij} + \delta_{kl}D_{ij}) + (\delta_{kl}D_{ij} + \delta_{ij}R_{kl})\right]$$
$$-\frac{1}{6}\left[\delta_{il}(R_{kj} + D_{kj}) - \delta_{kj}(R_{il} + D_{lj}) - \delta_{ik}(R_{lj} + D_{lj}) - \delta_{lj}(R_{ki} + D_{ki})\right] \tag{19}$$

$\mathbf{M}(\mathbf{R}, \mathbf{D}, \mathbf{Q}^h)$ can be modeled using $\mathbf{M}^h(\mathbf{R}^h, \mathbf{D}^h, \mathbf{Q}^h) \in \mathbf{Cov}(\mathbb{H}^2 \oplus \mathbb{H}^2 \oplus \mathbb{H}^3, \mathbb{H}^4)$. The difficulty in modeling $\mathbf{M}(\mathbf{R}, \mathbf{D}, \mathbf{Q}^h)$ is due to $\mathbf{Q}^h$ which is in $\mathbb{H}^3$. So, historically models were either ad-hoc (Launder et al. (1975)), only used a subset of tensor dependencies (Johansson & Hallbäck (1994) built $\mathbf{M}(\mathbf{R})$), or simply linear (Kassinos et al. (2001)). We present a complete $\mathbf{M}(\mathbf{R}, \mathbf{D}, \mathbf{Q}^h)$ model (using $\mathbf{M}^h$) non-linear up to degree 2 and 3 for the first time. The explicit models are shown in

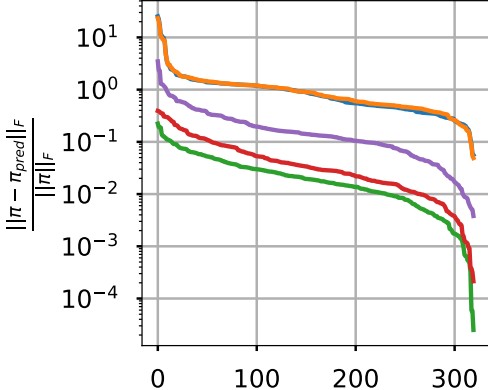

Figure 2: Error in the rapid term. Legend: —— Degree 3 model, —— Degree 2 model, —— Linear Model (Kassinos et al. (2001)), —— LRR Model (Launder et al. (1975)), —— IP Model (Launder (1989))

Appendix G.2. Computations show that a degree 2 $\mathbf{M}$ model can be built using 4 invariants and 6 tensor monomials and a degree 3 term is constructed with 11 Invariants and 27 tensor monomials. This is the underlying algebraic structure. For these two models, simple MLPs with 2 hidden layers with 128 neurons were used with input-output layer sizes of $(4, 6)$ and $(11, 27)$ respectively to learn coefficients for each tensor monomial as a function of the Invariants. In Fig. 2, these new models are compared against two classical LRR (Launder et al. (1975)) and IP (Launder (1989)), models and the linear $\mathbf{M}(\mathbf{R}, \mathbf{D}, \mathbf{Q}^*)$ model by Kassinos (1995). To learn the scalar coefficient functions for the representation functions of $\mathbf{M}$ tensor, 320 different flow scenarios were simulated using Rapid Distortion theory (Pope (2001)) and a dataset is curated with $\mathbf{M}, \mathbf{R}, \mathbf{D}, \mathbf{Q}^*$ tensors. A 75:20:5 split was used for train, validation and testing split, while ensuring that the testing data was the most anisotropic, which would make modeling most difficult. The results in Fig. 2 shows the error in modeling the $\mathbf{\Pi}$ tensor using different models. *The models we propose give – for the first time – an error which is lower than $10\%$ in most flow scenarios.* They perform *an order of magnitude* better than the linear model and almost *two orders of magnitude* better than the classical models.

## 6 REPRODUCIBILITY

All theoretical, computational, and numerical results in this paper are readily reproducible by the the methods described in the paper and/or by using the code included in out supplement. Specifically, three different cases were considered to showcase the applicability of the current work. In §4.1, and §4.2 the results were purely algebraic. For the first of these, the results do not require extensive computations. Following the second remark in the Appendix E, and using the covariant algebra in Table 2, the covariant module generators can be computed by hand. For the second case, we follow the same outline but use the covariant algebra in Table 3, but this needs explicitly invocation the Algorithm 1, because it has 4 invariants and Gröbner basis calculations are not trivial by hand. Source code is attached in the supplementary files, where this example is worked out end-to-end. In §5, an extensive numerical example is worked out. The primary result in minimal covariant module generators and the generators of the invariant algebra, which are summarized in Table 6 and Table 5. These results require covariant algebra generators of $S_6 \oplus S_4 \oplus S_4$ which is not readily available in literature. But, $S_6$ and $S_4 \oplus S_4$ are available separately. This can be used to create the joint covariant algebra by Gordan's algorithm as described in Appendix E, and then we proceed in the same way as before to identify the invariant generators and the covariant Module generators of $S_8$, In this case, we impose limits on the degrees to 2 and 3 to obtain quadratic and cubic models. Then, the covariants are converted back to tensors using Appendix B and trained using two simple feedforward neural networks whose descriptions are mentioned in Appendix G.2.

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

## A LARGE LANGUAGE MODEL USAGE

LLMs were not used to generate or research any parts of this paper.

## B TENSOR COVARIANTS

**Lemma B.1.** *Let $\boldsymbol{H}_1 \in \mathbb{H}^n(\mathbb{R}^3)$ and $\mathbf{H}_2 \in \mathbb{H}^p(\mathbb{R}^3)$ be two harmonic tensors. Let $r$ be the order of the transvectant, then*

$$\{\phi^* \mathbf{H}_1, \ \phi^* \mathbf{H}_2\}_{2r} = \frac{1}{2^r} \, \phi^*\big((\mathbf{H}_1{}^{(r)}\mathbf{H}_2)_0^s\big),$$

$$\{\phi^* \mathbf{H}_1, \ \phi^* \mathbf{H}_2\}_{2r+1} = \kappa(n,p,r) \, \phi^*\left((\operatorname{tr}^r(\mathbf{H}_1 \times \mathbf{H}_2))_0\right),$$

*where,*

$$\kappa(n,p,r) = \frac{1}{2^{2r+1}} \, \frac{(n+p-1)! \, (n-r-1)! \, (p-r-1)!}{(n+p-1-2r)! \, (n-1)! \, (p-1)!} \, .$$

*The tensor contraction is defined as*

$$(\mathbf{T}^{1\,(r)}\mathbf{T}^2)_{i_1 i_2 \ldots i_{p-r} j_{r+1} j_{r+1} \ldots j_q} = \delta_{i_{p-r+1} j_1} \ldots \delta_{i_p j_q} T^1_{i_1 i_2 \ldots i_p} T^2_{j_1 j_2 \ldots j_q},$$

*the symmetrization operation is*

$$\mathbf{T}^s = \frac{1}{n!} \sum_{\sigma \in \mathfrak{S}_n} T_{\sigma(1)\sigma(2)\ldots\sigma(n)},$$

*and $\mathbf{P}_0$ is the harmonic projection of the tensor $\mathbf{P}$. The generalized cross product between two totally symmetric tensors $\mathbf{S}^1 \in \mathbb{S}^p(\mathbb{R}^3)$ and $\mathbf{S}^2 \in \mathbb{S}^q(\mathbb{R}^3)$ is*

$$\mathbf{S}^1 \times \mathbf{S}^2 = (S^1 \times S^2)_{i_1 \ldots i_{p+1-1}} = (\epsilon_{i_1 jk} S^1_{j i_2 \cdots_p} S^2_{k i_{p+1} \ldots i_{p+q-1}})^s$$

*Finally, $\phi^* : \mathcal{H}_n(\mathbb{C}^3) \to S_{2n}$ is a unique equivariant isomorphism (up to a nonzero scale factor).*

## C HARMONIC DECOMPOSITION OF ANY TENSOR

From a representation theory perspective, the harmonic decomposition is an irreducible decomposition. One of the earliest works which showed harmonic decomposition is by Spencer (1970; 1987). A more refined version is developed by Zou et al. (2001), harmonic decomposition is referred to as deviatoric decomposition, and a recursive formula is shown which is particularly conducive for working with computer algebra software. Most works on harmonic decomposition occur in two steps. First, the tensor is decomposed into its symmetric components, then each symmetric component is decomposed into trace-less components. For example, a general order 2 tensor ($\mathbf{T} \in \mathbb{T}^2$) can be decomposed as,

$$T_{ij} = \delta_{ij} p + \epsilon_{ijk} u_k + b_{ij}$$

$$\text{where,} \quad p = \frac{1}{3} T_{kk}, \quad u_k = \frac{1}{2} \epsilon_{ijk} T_{jk}, \quad b_{ij} = \frac{1}{2}\left(T_{ij} + T_{ji}\right) - \frac{1}{3} \delta_{ij} T_{kk}. \tag{20}$$

Here, $\delta_{ij}$ is the kronocker-delta and $\epsilon_{ijk}$ is the alternating tensor. Here, $p \in \mathbb{H}^0$, $\mathbf{u} \in \mathbb{H}^1$, and $\mathbf{b} \in \mathbb{H}^2$ are irreducible representations. Outside of the complete harmonic decompision, the symmetric and traceless decompositions have also been studied. The theory of Young tableau (Fulton (1997)) can help with effective symmetric decomposition. More recently, the result given by Toth & Turyshev

(2022) can be adapted for the current context to begin with a symmetric tensor and decompose it into trace-less component. For $\mathbf{Q} \in \mathbb{S}\mathrm{ym}^k$, an iterative formula can be written as

$$\mathbf{Q}_0 = Q_{i_1 i_2 \cdots i_k} + \sum_{p=1}^{\lfloor k/2 \rfloor} (-1)^p \, \frac{k! \, (n + 2k - 2(p+2))!!}{2^p \, p! \, (k - 2p)! \, (n + 2k - 4)!!} \, \delta_{(i_1 i_2} \delta_{i_3 i_4} \cdots \delta_{i_{2p-1} i_{2p}} \, Q_{i_{2p+1} \cdots i_k)} \, ,$$

where the $n^{\text{th}}$ trace is defined as

$$Q_{i_1 i_2 \cdots i_{k-2n}} = \delta_{i_{k+1} i_{k+2}} \cdots \delta_{i_{k-1} i_k} Q_{i_1 i_2 \cdots i_k}.$$

Here $n$ is the number of dimensions, and $k$ is the order of the tensor $\mathbf{Q}$. Also,

$$a!! = \begin{cases} a(a-2)(a-4)\cdots 2, & \text{if } a \text{ is even,} \\ a(a-2)(a-4)\cdots 1, & \text{if } a \text{ is odd,} \end{cases} \qquad \text{with } 0!! = 1, \; 1!! = 1.$$

Using this formula, some common harmonic projection formulae are worked out below, for 3 dimensions. Here, $\mathbf{T}_n \in \mathbb{S}^n$ and $\mathbf{T}_n^0 \in \mathbb{H}^n$ is its corresponding harmonic projection.

$$\mathbf{T}_2^0 = T_{ij} - \frac{1}{3} \delta_{ij} T_{mm}$$

$$\mathbf{T}_4^0 = T_{ijkl} - \frac{6}{7} (\delta_{ij} T_{mmkl})^s + \frac{3}{35} (\delta_{ij} \delta_{kl})^s T_{mmnn}$$

$$\mathbf{T}_7^0 = T_{ijklmnp} - \frac{21}{13} (\delta_{ij} T_{qqklmnp})^s + \frac{105}{143} (\delta_{ij} \delta_{kl} T_{qqrrmnp})^s - \frac{105}{1287} (\delta_{ij} \delta_{kl} \delta_{mn} T_{qqrrttp})^s$$

## D    ISOMORPHISM BETWEEN BINARY FORMS AND HARMONIC TENSORS

The isomorphism which links $S_{2n}$ and $\mathbb{H}^n$ is crucial for the results established in §3. Let $\mathbf{T} \in \mathbb{H}^n[\mathbb{R}^3]$ which is an irreducible representation of SO(3, $\mathbb{R}$). It can be complexified with $\otimes\mathbb{C}$ to get $\mathbb{H}^n[\mathbb{C}^3]$, which is an irreducible representation of $SO(3, \mathbb{C})$. The *polarization* (see Appendix A in Olive et al. (2017)) map can be used to convert a order $n$ harmonic (or any symmetric) tensor into a homogeneous polynomial $p(x, y, z)$:

$$\psi : \mathbb{H}^n(\mathbb{C}^3) \to \mathbb{C}_n[\mathbb{C}^3]. \tag{21}$$

Here $\mathbb{C}_n[\mathbb{C}^3]$ is the space of homogeneous ternary polynomials (in 3 variables) of order $n$. This map is invertible so any polynomial $p(x, y, z)$ can also be mapped to a harmonic tensor[3]. There is an isomorphism between the terinary polynomial forms and binary forms which can be written using the Cartan map:

$$\mathbf{f}(u, v) = p\left( \frac{u^2 - v^2}{2}, \frac{u^2 + v^2}{2i}, uv \right), \tag{22}$$

where $\mathbf{f} \in S_{2n}$. The ternary form ($p \in \mathbb{C}_n[\mathbb{C}^3]$) can be recovered with the map below:

$$u^{2n-k} v^k \longrightarrow \begin{cases} z^k (x + iy)^{n-k}, & \text{if } 0 \le k \le n, \\ z^{2n-k} (-x + iy)^{k-n}, & \text{if } n \le k \le 2n. \end{cases}$$

This finishes the description of the isomorphism. The implementation of the isomorphism can be altered based on the application. For a more detailed treatment we refer to the works of Olive et al. (2017) and Olive et al. (2018).

---

[3]Now, if the tensor is symmetric and also trace free, then the laplacian of the corresponding homogeneous polynomial is zero, hence they are appropriately referred to as harmonic tensors.

# E   GORDAN'S ALGORITHM

Gordan's algorithm is used for generating the covariant algebra of binary forms. It was originally proposed Gordan (1868) first showed that the covariant algebra is finitely generated (even before Hilbert). Olive (2017) provides a reformulation of the algorithm using the language of modern representation and invariant theories. The Gordan's algorithm has two form. The first form is to derive $\mathbf{Cov}(S_n)$ using $\mathbf{Cov}(S_k)$ for k < n. The second form is to derive the covariant algebra of joint binary forms, i.e., $\mathbf{Cov}(S_{n_1} \oplus S_{n_2} \oplus \cdots \oplus S_{n_k})$. The Covariant algebra of simple binary forms is already derived for several orders. $S_5, S_6$ are derived by Gordan (1875), $S_7$ by Dixmier & Lazard (1985); Bedratyuk (2009), $S_8$ by Draisma (2014); Bedratyuk (2006), $S_9$ and $S_{10}$ by Lercier & Olive (2015). Using these equivariant tensor functions using dependencies upto $\mathbb{T}^5$ can be computed following the current work. Results for $\mathbf{Cov}(S_4)$, as an example, are shown in Table 2, where $\mathbf{v} \in S_4$.

| Degree / Order | 0 | 4 | 6 |
|---|---|---|---|
| 1 | | $\mathbf{v}$ | $(\mathbf{v}, (\mathbf{v}, \mathbf{v})_2)_1$ |
| 2 | $(\mathbf{v}, \mathbf{v})_4$ | $(\mathbf{v}, \mathbf{v})_2$ | |
| 3 | $(\mathbf{v}, (\mathbf{v}, \mathbf{v})_2)_4$ | | |

Table 2: Covariant Algebra of $S_4$ ($\mathbf{v} \in S_4$).

The second form of Gordan's algorithm uses the $\mathbf{Cov}(S_n)$ and $\mathbf{Cov}(S_m)$ to derive $\mathbf{Cov}(S_n \oplus S_m)$. If $\mathbf{f}_i \in S_n, \mathbf{g}_j \in S_m$ for $1 \le i \le p, 1 \le j \le q$ then let $a_i, b_j$ be the orders of $\mathbf{f}_i, \mathbf{g}_j$, a new transvectant can be constructed using $\alpha \in \mathbb{N}^p, \beta \in \mathbb{N}^q$ as $(\mathbf{f}_1^{\alpha_1} \mathbf{f}_2^{\alpha_2} \cdots \mathbf{f}_p^{\alpha_p}, \mathbf{g}_1^{\beta_1} \mathbf{g}_2^{\beta_2} \cdots \mathbf{g}_q^{\beta_q})_r$ for some $r$. Gordan's algorithm gives a finite set of $(\alpha, \beta, r)$s using which the covariant algebra can be generated. They are obtained as irreducible solutions (cannot be decomposed into sum of non-trivial solutions) of the linear Diophantine equation:

$$a_1\alpha_1 + a_2\alpha_2 + \cdots + a_p\alpha_p = u + r$$
$$b_1\beta_1 + b_2\beta_2 + \cdots + b_q\beta_q = v + r \tag{23}$$

| d/o | 0 | 2 | 4 | 6 |
|---|---|---|---|---|
| 1 | | | $\mathbf{p}, \mathbf{q}$ | |
| 2 | $(\mathbf{p}, \mathbf{p})_4, (\mathbf{q}, \mathbf{q})_4$ $(\mathbf{p}, \mathbf{q})_4$ | $(\mathbf{p}, \mathbf{q})_3$ | $(\mathbf{p}, \mathbf{p})_2, (\mathbf{q}, \mathbf{q})_2$ $(\mathbf{p}, \mathbf{q})_2$ | $(\mathbf{p}, \mathbf{q})_1$ |
| 3 | $(\mathbf{p}, (\mathbf{p}, \mathbf{p})_2)_4$ $(\mathbf{q}, (\mathbf{q}, \mathbf{q})_2)_4$ $(\mathbf{p}, (\mathbf{q}, \mathbf{q})_2)_4$ $(\mathbf{q}, (\mathbf{p}, \mathbf{p})_2)_4$ | $(\mathbf{p}, (\mathbf{q}, \mathbf{q})_2)_3$ $(\mathbf{q}, (\mathbf{p}, \mathbf{p})_2)_3$ | $(\mathbf{p}, (\mathbf{q}, \mathbf{q})_2)_2$ $(\mathbf{q}, (\mathbf{p}, \mathbf{p})_2)_2$ | $(\mathbf{p}, (\mathbf{p}, \mathbf{p})_2)_1$ $(\mathbf{q}, (\mathbf{q}, \mathbf{q})_2)_1$ $(\mathbf{p}, (\mathbf{q}, \mathbf{q})_2)_1$ $(\mathbf{q}, (\mathbf{p}, \mathbf{p})_2)_1$ |
| 4 | $((\mathbf{p}, \mathbf{p})_2, (\mathbf{q}, \mathbf{q})_2)_4$ | $((\mathbf{p}, \mathbf{p})_2, (\mathbf{q}, \mathbf{q})_2)_3$ $((\mathbf{p}, (\mathbf{p}, \mathbf{p})_2)_1, \mathbf{q})_4$ $(\mathbf{p}, (\mathbf{q}, (\mathbf{q}, \mathbf{q})_2)_1)_4$ | | |
| 5 | | $(\mathbf{p}^2, (\mathbf{q}, (\mathbf{q}, \mathbf{q})_2)_1)_6$ $((\mathbf{p}, (\mathbf{p}, \mathbf{p})_2)_1, \mathbf{q}^2)_6$ | | |

Table 3: Covariant Algebra of $S_4 \oplus S_4$ ($\mathbf{p}, \mathbf{q} \in S_4$)

Using this algorithm, the $\mathbf{Cov}(S_6 \oplus S_2)$, $\mathbf{Cov}(S_4 \oplus S_4)$, $\mathbf{Cov}(S_6 \oplus S_4)$, $\mathbf{Cov}(S_6 \oplus S_4 \oplus S_2)$ are derived by Olive (2017). $\mathbf{Cov}(S_4 \oplus S_4)$ is shown in Table 3 for reference, by ordering them degree and order wise. Here, $\mathbf{p} \in S_4, \mathbf{q} \in S_4$.

**Remark.** *Solutions to linear Diophantine equations can become expensive when the number of variables is large. In the current work, the Gordan's algorithm and also numerical polynomial remainder algorithm proposed in Appendix F can involve linear Diophantine equations with hundreds (if not thousands) of variables. In practice, the fastest strategy to obtain solutions is by first creating a convex cone and computing its lattice points. The algorithms described in the Normaliz (Bruns et al. (2017)) Package are recommended.*

**Remark.** *It should be noted that the Gordan's algorithm attempts to produce a finite covariant algebra and not a finite covariant module basis which is what we need. So, after the algebra (for example, $V$) is generated by Gordan's algorithm, the covariant module generators of $\mathbf{Cov_G}(\mathbf{V}, \mathbf{S_n})$ are generated by multiplying the polynomials in the algebra together with one another such that they match the order of $\mathbf{S}_n$ and some degree. This process can be repeated degree-per-degree to find covariant generators in each degree. Since, the algebra is finitely generated, after some degree, new generators will not be identified. This problem of finding all possible combinations of polynomial multiplication for a particular degree and order can also be achieved by solving a linear Diophantine equation.*

## F  POLYNOMIAL REMAINDER

In Algorithm 15, the operation $f_i\%\mathfrak{m}$ involves calculating the polynomial remainder after dividing by an ideal. This can be significantly expensive. It is convenent to use Gröbner basis calculations. In Macaulay2, creating an Ideal pre-computes the Gröbner basis. Also, since the ideal $\mathfrak{m}$ is homogeneous, degree limits can be imposed on S-pairs that are used in the basis computations, which tremendously reduces the expenditure. However, if the maximum degree under consideration is too high, the calculation of Gröbner basis might be too expensive. So, a numerical linear algebra based algorithm is desirable to avoid symbolic computations altogether.

Instead of computing the polynomial reminders for every generator and then computing the linearly independence, the linear independence test can be done in a degree by degree, order by order manner. An intermediate degree, order $(d, o)$ stage computations would involve the following steps. Let $\mathcal{H}_{(d,o)}$ be the non-minimal module generators who have a degree $d$, and order $o$. Let the $R$ linearly independent module generators identified up to $(d, o)$ be $\mathcal{H}^{\mathrm{LI}}_{(<d,<o)}$. By multiplying polynomials in $\mathcal{H}^{\mathrm{LI}}_{(<d,<o)}$ together among themselves and using the generators of $R$, the $(d, o)$ piece can be spanned, by solving an appropriate linear Diophantine equation for the degree and order. The polynomials in $\mathcal{H}_{(d,o)}$ which are linearly among themselves and also linearly independent with this newly spanned set can be added to $\mathcal{H}^{\mathrm{LI}}_{(<d,<o)}$ to complete the calculation for the degree, order $(d, o)$ pair. Next, the calculation could focus either on $(d + 1, o)$ or $(d, o + 1)$ piece until the module generators are exhausted.

## G  MINIMAL ANVARIANT ALGEBRA AND COVARIANT MODULE GENERATORS

### G.1  MODELING A SYMMETRIC ORDER 2 TENSOR USING TWO SYMMETRIC ORDER 2 TENSORS

| Deg | $2\mathbb{H}^2$ | $2\mathbb{H}^0$ |
|---|---|---|
| 1 | | $\mathrm{tr}(\mathbf{B}), \mathrm{tr}(\mathbf{C})$ |
| 2 | $\mathrm{tr}(\mathbf{B}^2), \mathrm{tr}(\mathbf{C}^2), \mathrm{tr}(\mathbf{BC} + \mathbf{CB})$ | |
| 3 | $\mathrm{tr}(\mathbf{CH}_{2c} + \mathbf{H}_{2c}\mathbf{C}), \mathrm{tr}(\mathbf{BH}_{2b} + \mathbf{H}_{2b}\mathbf{B}), \mathrm{tr}(\mathbf{CH}_{2b} + \mathbf{H}_{2b}\mathbf{C}), \mathrm{tr}(\mathbf{BH}_{2c} + \mathbf{H}_{2c}\mathbf{B})$ | |
| 4 | $\mathrm{tr}(\mathbf{H}_{2c}\mathbf{H}_{2b} + \mathbf{H}_{2b}\mathbf{H}_{2c})$ | |

Table 4: Invariants of $2\mathbb{H}^2 \oplus 2\mathbb{H}^0$.

### G.2 Non-Linear Rapid Pressure Strain Rate tensor models

To model the $\mathbf{\Pi}$ tensor, the order $4$ $\mathbf{M}$ tensor must be modeled. Eq. 19 shows this requires modeling a $\mathbb{H}^4$ using $\mathbb{H}^2 \oplus \mathbb{H}^2 \oplus \mathbb{H}^3$. This requires identifying the minimal generators of the order $8$ covariant module of $S_6 \oplus S_4 \oplus S_4$. Using the Covariant Algebra of $S_6$ and $S_4 \oplus S_4$ (Olive (2017)), Gordan's algorithm can be used to get the module generators of $S_6 \oplus S_4 \oplus S_4$. From those the order $8$ and degree $\leq 3$ covariant modules generators are identified and minimized (using Algorithm 1). There are $4$ degree $2$ invariants and $7$ degree $3$ invariants which are shown in Table 6. There are $6$ degree $2$ and $21$ degree $3$ minimal covariant generators shown in Table 5, which will be converted to tensor monomials. The coefficients to these monomials will be learned as scalar functions of invariants, which will be learned using neural networks. The coefficients of the degree $2$ and degree $3$ models are leaned using two neural networks with similar architectures. The networks have two hidden layers with of $128$ neurons and have 18k and 20k parameters respectively. We used AdamW for regularization and dropout was used for regularization.

| d/o | 8 |
|---|---|
| 2 | $(\mathbf{f}, \mathbf{p})_1, (\mathbf{f}, \mathbf{q})_1, (\mathbf{f}, \mathbf{f})_2, \mathbf{p}^2, (\mathbf{q}, \mathbf{p})_0, \mathbf{q}^2$ |
| 3 | $(\mathbf{f}, (\mathbf{p}, \mathbf{p})_2)_1, (\mathbf{f}, (\mathbf{q}, \mathbf{q})_2)_1, (\mathbf{f}, (\mathbf{p}, \mathbf{q})_2)_1, (\mathbf{f}, (\mathbf{p}, \mathbf{q})_1)_2, (\mathbf{f}, (\mathbf{f}, \mathbf{p})_4)_0, (\mathbf{f}, (\mathbf{f}, \mathbf{q})_4)_0,$ $((\mathbf{f}, \mathbf{f})_2, \mathbf{p})_2, ((\mathbf{f}, \mathbf{f})_2, \mathbf{q})_2, ((\mathbf{f}, \mathbf{f})_4, \mathbf{f})_1, (\mathbf{p}, (\mathbf{f}, \mathbf{f})_4)_0, (\mathbf{p}, (\mathbf{f}, \mathbf{p})_3)_0, (\mathbf{p}, (\mathbf{f}, \mathbf{q})_3)_0,$ $(\mathbf{q}, (\mathbf{f}, \mathbf{f})_4)_0, (\mathbf{q}, (\mathbf{f}, \mathbf{p})_3)_0, (\mathbf{q}, (\mathbf{f}, \mathbf{q})_3)_0, ((\mathbf{p}, \mathbf{p})_2, \mathbf{p})_0, ((\mathbf{p}, \mathbf{p})_2, \mathbf{q})_0, ((\mathbf{q}, \mathbf{q})_2, \mathbf{p})_0,$ $((\mathbf{q}, \mathbf{q})_2, \mathbf{q})_0, ((\mathbf{p}, \mathbf{q})_2, \mathbf{p})_0, ((\mathbf{p}, \mathbf{q})_2, \mathbf{q})_0$ |

Table 5: Order $8$ Covariant module generators of $S_6 \oplus S_4 \oplus S_4$ up to degree 3. ($\mathbf{p}, \mathbf{q} \in S_4, \mathbf{f} \in S_6$)

| d/o | 0 |
|---|---|
| 2 | $(\mathbf{f}, \mathbf{f})_6, (\mathbf{p}, \mathbf{p})_4, (\mathbf{q}, \mathbf{q})_4, (\mathbf{p}, \mathbf{q})_4$ |
| 3 | $(\mathbf{p}, (\mathbf{p}, \mathbf{p})_2)_4, (\mathbf{q}, (\mathbf{q}, \mathbf{q})_2)_4, (\mathbf{p}, (\mathbf{q}, \mathbf{q})_2)_4, (\mathbf{q}, (\mathbf{p}, \mathbf{p})_2)_4,$ $(\mathbf{f}, (\mathbf{p}, \mathbf{q})_1)_6, ((\mathbf{f}, \mathbf{f})_4, \mathbf{q})_4, ((\mathbf{f}, \mathbf{f})_4, \mathbf{p})_4$ |

Table 6: Invariant ring generators of $S_6 \oplus S_4 \oplus S_4$ up to degree 3. ($\mathbf{p}, \mathbf{q} \in S_4, \mathbf{f} \in S_6$)

## H Overview

The Fig. 3 shows the overall workflow for learning function representations developed in this work.

## I Extensions to other groups

The following general formula is used in the current work:

$$\mathbf{T}(\mathbf{A}_1, \mathbf{A}_2, \cdots, \mathbf{A}_k) = \sum_{i=1}^{\gamma} C_i(\lambda_1, \lambda_2, \cdots, \lambda_m) \mathbf{B}_i(\mathbf{A}_1, \mathbf{A}_2, \cdots, \mathbf{A}_k)$$

For mappings between any representation of any group $G$, this ansatz holds. The generators for the Invariant ring and the Covariant module over that ring are necessary for building functions between representations of any general group. In this work tensor representations over $SO(3)$ were considered, and algorithms were developed for identifying generators of Invariants and Covariant modules. For extension of this work to other representations of different groups, appropriate algorithms for generators must be identified, which can then be used with Algorithm 1 to build the representation maps. There is extensive literature for efficient computation of invariant rings and covariant module generators (Derksen & Kemper (2015); Goodman et al. (2009)).

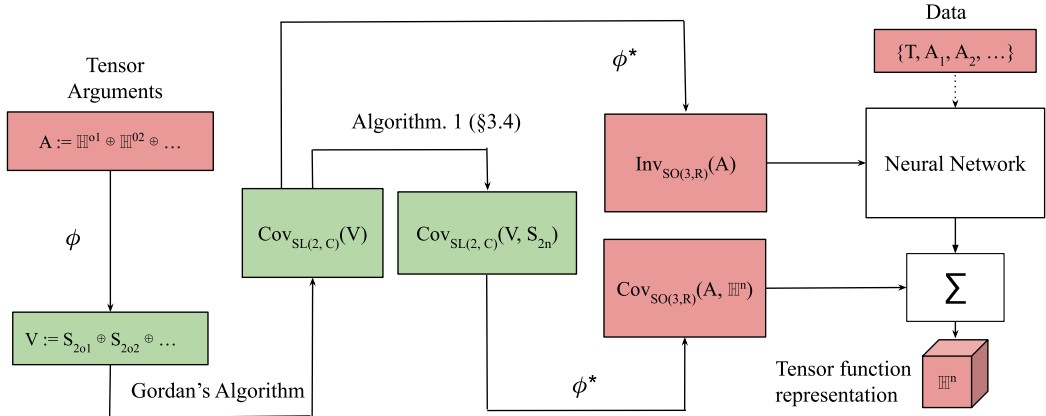

Figure 3: Workflow for learning tensor function representations of order $n$ harmonic tensor $\mathbf{T}$ using $(\mathbb{H}^{o1} \oplus \mathbb{H}^{o2} \oplus \cdots)$. Red blocks work with tensors, and green blocks work with binary forms. $\phi$ is the isomorphism between them and $\phi^*$ is the inverse map of the isomorphism.

Let $\mathcal{V} := \mathbb{T}^{n_1} \oplus \mathbb{T}^{n_2} \oplus \cdots \mathbb{T}^{n_k}$ be a tensor space. Then the G-equivariant tensor function representations over $\mathcal{T} := \mathbb{T}^t$ can be derived as covariant module, $\mathrm{Cov}_G(\mathcal{T})$ generators over the Invariant ring $\mathrm{Inv}_G(\mathcal{V})$. This work focused on the $G = \mathrm{SO}(3, \mathbb{R})$ extensively, but the results can be extended to the $\mathrm{O}(3, \mathbb{R})$ in a straight forward manner.

## I.1 $\mathrm{O}(3, \mathbb{R})$-EQUIVARIANT TENSOR FUNCTIONS

The $\mathrm{O}(3, \mathbb{R})$ group contains both reflections and rotations, unlike $\mathrm{SO}(3, \mathbb{R})$ which only contains proper rotation matrices ($\mathrm{SO}(3) \subset \mathrm{O}(3)$). Since, $\mathrm{O}(3)$ is a bigger group, $\mathrm{O}(3)$-equivariance is more restrictive than $\mathrm{SO}(3)$-equivariance. Once, the $\mathrm{SO}(3)$-equivariant functions are derived using the Gordan's algorithm, the following Lemma can be used to filter out the $\mathrm{O}(3)$-equivariant tensor functions.

**Lemma I.1.** *Let $\mathbf{T}(\mathbf{A}_1, \mathbf{A}_2, \cdots, \mathbf{A}_k)$ be a tensor function of order, o($\mathbf{T}$) and degree, d($\mathbf{T}$), then $\mathbf{T}$ is an O(3) equivariant function representation, iff:*

$$o(\mathbf{T}) \cong \sum_{i=1}^{n} o(\mathbf{A_i})d(\mathbf{A_i}) \quad + d(\boldsymbol{\epsilon}) \quad (mod \quad 2) \tag{24}$$

Where, $d(\boldsymbol{\epsilon})$ is degree (number of copies) of levi-civita tensors in $\mathbf{T}$. In its general form, the expression takes the following form including the degree of kronocker-deltas, $d(\boldsymbol{\delta})$ as:

$$o(\mathbf{T}) \cong \sum_{i=1}^{n} o(\mathbf{A_i})d(\mathbf{A_i}) \quad + o(\boldsymbol{\delta})d(\boldsymbol{\delta}) + \quad o(\boldsymbol{\epsilon})d(\boldsymbol{\epsilon}) \quad (mod \quad 2)$$

$$o(\mathbf{T}) \cong \sum_{i=1}^{n} o(\mathbf{A_i})d(\mathbf{A_i}) \quad + 2d(\boldsymbol{\delta}) + \quad 3d(\boldsymbol{\epsilon}) \quad (mod \quad 2)$$

$$o(\mathbf{T}) \cong \sum_{i=1}^{n} o(\mathbf{A_i})d(\mathbf{A_i}) + \quad d(\boldsymbol{\epsilon}) \quad (mod \quad 2)$$

## I.2 PERMUTATION EQUIVARIANT TENSOR FUNCTIONS

As noted in Col. 3.1.1 and in Fig. 3, the co-domain is decomposed into harmonic tensor spaces using harmonic decomposition. Upon plugging in the models of the harmonic components back into the decomposition the tensor function representations of the desired space are build by keeping the correct symmetries of the group. So, it is worthwhile to mention that the current methodology in

this work implicitly imposes equivariance of the permutation group through the harmonic decomposition. Extensions to groups in higher dimensions ($> 3$) and other general groups are out of scope for this current work.

## J    MORE MATHEMATICAL BACKGROUND

Some mathematical background and preliminaries used throughout this work were defined in Section. 2. Here, more prerequisites are presented with brief examples.

A *ring $R$* is a set accompanied with the addition and multiplication operations such that $R$ is an abelian group with respect to addition ($0 \in R$, if $x \in R$, then $-x \in R$) and the multiplication is associative and distributive over addition. If a ring is *commutative*, $xy = yx, \forall x, y \in R$. An example is the *polynomial ring* which is commonly encountered this work. The set of all polynomials

$$f(x) = a_0 + a_1 x + \cdots + a_n x^n$$

for some $n > 0$ and $a_i \in R$, where $x$ is an indeterminate, form the polynomial ring $R[x]$ with coefficients in $R$.

If $R$ is a ring, then an *R-module* is an abelian group $M$ equipped with a multiplication by elements of $R$, such that :

$$a(x + y) = ax + ay$$
$$(a + b)x = ax + bx$$
$$(ab)x = a(bx)$$
$$1x = x$$

for all $a, b \in R, x, y \in M$. If $u_1, u_2, \ldots u_n \in M$ are such that all elements of M can be written as $x_1 u_1 + x_2 u_2 + \ldots x_n u_n$ for some $x_i \in R$, then $u_1, u_2, \ldots u_n \in M$ is called as the *generating set* of $M$. It is called *minimal* if removing any $u_i$ would disable the generating set from spanning the entire module. It is called as a *basis* if the elements are $R$-linearly independent. It is possible to have minimal generating sets of different dimensions. For example on $\mathbb{Z}$, both $\{1\}$ and $\{2, 3\}$ are minimal generators. But only $\{1\}$ is a basis.

