# OpenReview forum: "Learning equivariant tensor function representations via covariant algebra of binary forms"
_ICLR.cc/2026/Conference — Submitted to ICLR 2026_

### Official Review · Reviewer_sKrU · 2025-10-24

**Soundness:** 4
**Presentation:** 2
**Contribution:** 4
**Rating:** 8
**Confidence:** 4

**Summary:**

The authors look to establish a framework for creating SO(3,R) equivariant models from tuples of tensors to a tensor output that is based on two separate parts. The first part comes from deriving the algebraic structure (a complete and minimal generating set of tensor monomials) using ideas from classical invariant theory. The second part uses neural networks on domain-specific data (from problems in physics) to learn certain scalars that results in a model that is consistent with the physics (equivariant to SO(3,R)).

**Strengths:**

I would like to commend the authors on producing a strong piece of work. Particular strengths I see include:

- the use of classical invariant theory to pose their problem in terms of two separate parts: the ability to build the tensor functions out of minimal generators for binary forms using an isomorphism via harmonic tensors is very nice, as well as using neural nets to learn the physics (the scalars). I think it is clever to create models that have a fundamental algebraic structure underneath them so that the overall model is grounded in solid mathematical theory, with the neural nets essentially tweaking the model to the problem at hand.
- the creation of a novel algorithm to obtain minimal generators for Inv_G and Cov_G.
- the demonstration of their framework on a practical problem, obtaining SOTA results.

**Weaknesses:**

I also think there are some weaknesses in the paper, mostly relating to clarity of presentation, hence my score above. I think

- given that the mathematical theory is "advanced" and may not be very well known, the presentation given here, in my view, is far too "slick". I strongly recommend that the authors include, in the Appendix, a full background of definitions used in Section 2 (from Definition 2.2 onwards). Examples for each definition would also be very welcome, otherwise the paper is in danger of being intractable to the uninitiated.
- it took me a while to understand, from their introduction, what problem they were actually trying to solve - I think this should be made much clearer. One thing that might be helpful is to give the domain and codomain of each of the functions involved in (1), at worst in the Appendix. I'd even reorder the paragraphs in the Introduction to 1) state what has been done before *first* before saying 2) what their approach is to tackle the problem, otherwise it isn't obvious what the issues are, and their relevance (in relation to what has gone before). I did get there in the end, though.
- I also think the authors need to reread the paper for typos, lack of spaces between words, full stops coming before a reference (e.g Table. 2), consistency (e.g equation vs Equation vs Equ. vs (Equ)), and also some poor grammar (e.g lines 273-282). I appreciate for the last one that the authors may not be native English speakers but I felt that all of these things detracted a bit from the overall presentation. It would be a shame if these weren't fixed in a camera ready version since the idea itself is particularly brilliant.

**Questions:**

1) Does the framework only work for SO(3, R), or could it be extended to other groups beyond SO(3,R)? How general could the framework be?
2) What is the runtime of the Algorithm 1 that is presented? It seems to me like there may be scenarios where this wouldn't work (if the invariant ring is too large), but this is an opinion I loosely hold, so I would welcome any comments from the authors on this.
3) How does your approach compare to something like Villar et al. 2022 (Scalars are universal: Equivariant machine learning, structured like classical physics) which finds universally approximating polynomial functions to learn equivariant functions (by learning scalars)? I would suggest it might even fall under related work.

Finally a minor point: in definition 2.3 shouldn't it be n choose i not n-i choose i?

---

> ### Author Response · Authors · 2025-12-03
> **Response to Reviewer sKrU**
>
> ## **Addressing Weaknesses**
> > - given that the mathematical theory is "advanced" and may not be very well known, the presentation given here, in my view, is far too "slick". I strongly recommend that the authors include, in the Appendix, a full background of definitions used in Section 2 (from Definition 2.2 onwards). Examples for each definition would also be very welcome, otherwise the paper is in danger of being intractable to the uninitiated.
>
> Thank you for pointing this out. We have expanded the appendix to clarify the relevant definitions and now include explicit explanations of rings, modules, algebras, generating sets, and minimal generating sets, along with new illustrative examples for each. Please refer to the newly added Appendix J for these details.
>
> > - it took me a while to understand, from their introduction, what problem they were actually trying to solve - I think this should be made much clearer. One thing that might be helpful is to give the domain and codomain of each of the functions involved in (1), at worst in the Appendix. I'd even reorder the paragraphs in the Introduction to 1) state what has been done before first before saying 2) what their approach is to tackle the problem, otherwise it isn't obvious what the issues are, and their relevance (in relation to what has gone before). I did get there in the end, though.
>
> To address this issue, we have rewritten the introduction and revised the statement of contributions so that the problem and its motivations are presented more directly. We have also added the domain and codomain for each function in equation (1) in the Introduction section.
>
> > - I also think the authors need to reread the paper for typos, lack of spaces between words, full stops coming before a reference (e.g Table. 2), consistency (e.g equation vs Equation vs Equ. vs (Equ)), and also some poor grammar (e.g lines 273-282).
>
> We have carefully reviewed the manuscript and corrected typos, inconsistent abbreviations, and grammatical issues, and we will continue refining the draft to ensure a clean and consistent camera ready version.

---

> ### Author Response · Authors · 2025-12-03
> **Response to Reviewer sKrU (continued)**
>
> ## **Responses to Questions**
>
> > 1. Does the framework only work for SO(3, R), or could it be extended to other groups beyond SO(3,R)? How general could the framework be?
>
> We address the generality of our framework in detail in the newly added Appendix J. In summary, our ansatz applies to representation maps of arbitrary groups. However, suitable algorithms are required to identify generators of the invariant ring and the corresponding covariant module for each case. We demonstrate this concretely for SO(3), and we have now added extensions for O(3) and permutation-group equivariance in Appendix J.
>
> > 2. What is the runtime of the Algorithm 1 that is presented? It seems to me like there may be scenarios where this wouldn't work (if the invariant ring is too large), but this is an opinion I loosely hold, so I would welcome any comments from the authors on this.
>
> Thank you for raising this concern. The non linear models mentioned in Section 5 took a few minutes for completing the entire pipeline of generating the invariant rings and covariant module from gordans algorithm, and then applying algorithm 1.
> However, there are scenarios in which the computations may become expensive. Below, we provide several guidelines (also mentioned in the paper in various locations) to ensure scalability:
>
> 1) If the invariant ring is large, Gröbner basis computations may become expensive. We recommend imposing suitable degree limits on the S-pairs considered during the basis computation to control complexity. Alternatively, the numerical algorithm introduced in Appendix F provides a practical approach that avoids Gröbner basis computations altogether.
> 2) Computing function representations at higher degrees may become disproportionately expensive relative to the potential accuracy gains. We therefore recommend proceeding degree by degree and selecting an appropriate degree cutoff based on the application (Section 3.3).
> 3) Using modular arithmetic for Gaussian elimination with a large prime such as 65,521 (the largest prime less than $2^{16}$) can substantially accelerate the computations (Section 3.3).
> 4) Using the Normaliz package, lattice points of a cone can be used to solve large linear Diophantine systems, which standard tools (e.g., SciPy) are often unable to handle at scale (Remark 1 in Appendix E).
>
>
>
> > 3. How does your approach compare to something like Villar et al. 2022 (Scalars are universal: Equivariant machine learning, structured like classical physics) which finds universally approximating polynomial functions to learn equivariant functions (by learning scalars)? I would suggest it might even fall under related work.
>
> Thanks for bringing this to our attention. This paper closely follows the ansatz of our work and develops equivariant functions parametrized purely using scalars. We view the current work as enriching the methods developed in [1]. While [1] proposes method focusing on vector inputs, we show explicit methods to handle tensor inputs and tensor outputs directly by finitely generating the invariant ring and the corresponding covariant module. It would be interesting to see how the Algorithm 1 developed in our work could be applied to the models developed in [1]. We also evaluated our approach on the toy O(3)-equivariance task from [1] and observed that, given sufficient training data, our method achieves comparable performance while using simple MLPs with significantly smaller capacity.
>
> > Finally a minor point: in definition 2.3 shouldn't it be n choose i not n-i choose i?
>
> This typo has now been rectified in the manuscript.
>
> [ 1 ] Villar, Soledad, et al. ”Scalars are universal: Equivariant machine learn-ing, structured like classical physics.” Advances in neural information processing
> systems 34 (2021): 28848-28863.

---

### Official Review · Reviewer_JCzB · 2025-10-28

**Soundness:** 3
**Presentation:** 3
**Contribution:** 3
**Rating:** 4
**Confidence:** 2

**Summary:**

This paper presents a hybrid framework for learning equivariant tensor-valued functions, which are common in physical sciences like fluid dynamics. The core idea is to decompose the modeling problem into two parts: 1) an analytical part that uses classical invariant theory (specifically, the algebra of binary forms) to a priori determine a minimal and complete set of equivariant basis tensors (monomials) and scalar invariants; and 2) a learning part where a neural network is used to learn the coefficients for these basis tensors as functions of the invariants.The authors claim that previous attempts at this were computationally intractable for complex problems. Their key technical contribution is a set of "numerically efficient algorithms" (notably Algorithm 1) that can find this minimal basis, overcoming this limitation. The framework is validated in two ways: first, by correctly reproducing known classical results for simpler tensor functions, and second, by applying it to a "well-known bottleneck" in turbulence modeling (the RSPR tensor) and achieving results one to two orders of magnitude better than existing models.

**Strengths:**

1. The paper’s main strength is its hybrid design, which combines the rigor of analytical methods (guaranteed equivariance, interpretability, and a provably minimal basis) with the flexibility of neural networks (learning complex, non-linear relationships from data).
2. The application to the RSPR turbulence model is a strong, non-trivial test case. The proposed models achieve a one to two order-of-magnitude error reduction over established models, a highly significant improvement on a real-world problem.
3. The authors wisely validate their framework by first reproducing known, classical results from the literature (Section 4). This is a commendable step that builds significant trust in their complex mathematical machinery.
4. The paper claims to solve a long-standing computational bottleneck in this field. The development of efficient algorithms (like Algorithm 1) to find minimal generators is a key technical contribution that enables this framework to be applied to problems of practical interest

**Weaknesses:**

The paper claims its models are "economical" and that standard Equivariant Neural Networks (ENNs) are "computationally expensive". However, these claims are not substantiated with a direct quantitative comparison. The paper does not compare against an ENN baseline in terms of parameter count, training time, or inference cost for the RSPR problem. This makes it difficult to assess the full practical advantage of this method beyond its impressive accuracy.

**Questions:**

To help substantiate the claims of an "economical" model, could the authors provide a quantitative comparison of their final model's complexity? Specifically, what is the parameter count of the coefficient-learning neural network for the Degree 3 RSPR model, and how does this compare to a standard ENN baseline designed for the same task?

---

> ### Author Response · Authors · 2025-12-03
> **Response to Reviewer JCzB**
>
> ___
> ## **Responses to Questions**
>
> >To help substantiate the claims of an "economical" model, could the authors provide a quantitative comparison of their final model's complexity? Specifically, what is the parameter count of the coefficient-learning neural network for the Degree 3 RSPR model, and how does this compare to a standard ENN baseline designed for the same task?
>
> Thank you for raising this question. We have added additional details about model capacity in Section 5. In summary, for both the degree-2 and degree-3 models of the order-4
> M-tensor, we used a simple MLP with two hidden layers of size 128 which accounts to approximately 20k learnable parameters. While fine-tuning the architecture or using more complex models could potentially yield further improvements, such exploration is beyond the scope of the present work.
>
> Regarding comparisons with ENNs, we similarly consider this outside the scope of the current study. Our claims about model economy are grounded in the minimality of the underlying algebraic structure that our framework constructs.
> Given that we use minimal generators, with theoretically minimum number of terms,  it is likely plausible that our models would require fewer trainable parameters and simpler architectures. However, this claim remains unsubstaniated, so we have revised the Introduction section to reflect this. Now, we simply introduce our work as an algebraically minimal, more interpretable alternative to black box ENN architectures.

---

### Official Review · Reviewer_o5pb · 2025-10-29

**Soundness:** 3
**Presentation:** 1
**Contribution:** 3
**Rating:** 6
**Confidence:** 2

**Summary:**

This paper proposes a mathematically interpretable framework for equivariant tensor function representation.
The core idea is to establish a connection between harmonic tensors and binary forms (homogeneous polynomials), which enables the efficient identification of a minimal generating set for tensor function spaces using algebraic tools derived from Nakayama’s Lemma.
The authors further demonstrate the approach through numerical experiments, including both illustrative examples and a more practical case study on turbulence modeling.

**Strengths:**

1. The paper provides a clear and mathematically rigorous contribution, offering an interpretable method to derive minimally complete tensor function representations under symmetry constraints.
2. The work broadens the application scope of machine learning by bridging classical invariant theory and modern data-driven modeling, enriching cross-disciplinary understanding.

**Weaknesses:**

The main weakness is readability.
While the mathematical formulation is elegant, the presentation may be difficult to follow for readers without a strong background in algebraic geometry or invariant theory.
This is not a major flaw but may limit accessibility for the broader ML community.
I do not have strong critical concerns—mostly clarifying questions.

I am inclined to give a borderline positive score, primarily due to the novel connection established between harmonic tensors and binary forms (assuming this is the first work to do so). However, my final evaluation will depend on the authors’ responses, particularly regarding the clarity and readability of the presentation.

**Questions:**

1. On Eq. (1): Could you clarify the definition of “joint invariants” \lambda?
Specifically, what does the phrase “to express any other joint invariant” refer to?
Does it mean that all scalar invariants of multiple tensors can be expressed as functions of these \lambda_i’s?
2. Importance of minimal representation:
Could you elaborate on why finding a minimal representation is crucial?
Does it primarily improve computational efficiency, interpretability, generalization, or reduce overfitting?
Some concrete justification would help.
3. Notation clarity:

	•	Line 113: Please define GL(W) explicitly (general linear group of W?). This notation might not be immediately clear to ML readers.

	•	Line 124: Does “K-representations” refer to representations over a field K?

	•	Line 145: The symbol V should be bold, consistent with Eq. (6).

	•	Line 162: The notation SL should be in italics to maintain consistency.

4. Clarifications on algebraic statements (around line 196):
It would strengthen the paper to provide brief justification or reference for the following key claims:

	•	“\lambda_i’s must generate the invariant algebra…”

	•	“If the \lambda_i’s and B_i’s are minimal generators, then…”

	•	“Once the model is developed, the coefficient functions C_i’s must be learned….”

5. On the neural network implementation:
Could you discuss how factors such as the number of layers, neurons, initialization, or the scale of training data influence results?
Do different architectures affect the learned coefficient functions C_i(\lambda)?

6. On non-uniqueness:
Since neural networks typically do not yield unique solutions due to nonconvex optimization, how does this non-uniqueness or local convergence behavior affect the learned tensor function representation?

7. Minor issue:
In line 417, there is a missing blank space.

__Disclosure of AI Assistance__

Portions of this review (including wording refinement, grammar correction, and structural organization) were assisted by OpenAI’s GPT-5 language model.
The content, opinions, and evaluations are entirely my own, while the language model was used solely for improving clarity, coherence, and presentation.

---

> ### Author Response · Authors · 2025-12-03
> **Response to Reviewer o5pb**
>
> ## **Addressing Weaknesses**
>
> > 1. The main weakness is readability. While the mathematical formulation is elegant, the presentation may be difficult to follow for readers without a strong background in algebraic geometry or invariant theory. This is not a major flaw but may limit accessibility for the broader ML community. I do not have strong critical concerns—mostly clarifying questions.
>
> We thank the reviewer for raising this issue. To improve clarity several sections of the paper have been rewritten and highlighted to note the changes. We have also added new sections in the appendix to add more pre-requisites and mathematical background to make the subject more accessible to the ML community.

---

> ### Author Response · Authors · 2025-12-03
> **Response to Reviewer o5pb (continued)**
>
> ## **Responses to Questions**
>
> >1. On Eq. (1): Could you clarify the definition of “joint invariants” \lambda? Specifically, what does the phrase “to express any other joint invariant” refer to? Does it mean that all scalar invariants of multiple tensors can be expressed as functions of these \lambda_i’s?
>
> Yes, the set of $\lambda$ s is the generating set of the Invariant ring. So, any other scalar invariant can be expressed using the generators. We have re-expressed this in introduction for better clarity.
>
> >2. Importance of minimal representation: Could you elaborate on why finding a minimal representation is crucial? Does it primarily improve computational efficiency, interpretability, generalization, or reduce overfitting? Some concrete justification would help.
>
> In general, working with minimal sets of invariant and equivariant tensor functions reduces the number of terms involved by several orders of magnitude. For example, Gordans algorithm gives 695,754 invariants for Inv(S8 + S4 + S4) which forms a generating set. Out of these, only 287 invariants are sufficient to form a minimal generating set [1].
>
> In the paper, our methodology derives the minimal generators of invariants and covariant modules in a degree per degree manner. Using minimal representations is central to the overall computational tractability of the modeling problem. Working with minimally complete models improves interpretability by isolating specific tensor monomials which drive the desired behavior rather than obscuring it with a large number of terms. Regarding generality, it is mathematically equivalant to use any generating set or a minimal one. But the minimal generating set would require lesser number of terms and hence lesser learnable parameters and ultimately lesser training data. We also clarified the motivation for using minimal representations in the Introduction section.
>
> [ 1 ] Olive, Marc. "About Gordan’s algorithm for binary forms." Foundations of Computational Mathematics 17.6 (2017): 1407-1466.
>
> >3. Notation clarity:
> >- Line 113: Please define GL(W) explicitly (general linear group of W?). This notation might not be immediately clear to ML readers.
> >- Line 124: Does “K-representations” refer to representations over a field K?
> >- Line 145: The symbol V should be bold, consistent with Eq. (6).
> >- Line 162: The notation SL should be in italics to maintain consistency.
>
> Thank you for pointing it out. We are indeed referring to the General Linear group as you suggested. It has been explicitly defined now for clarity. The other typos you mentioned were also rectified.
>
> >4. Clarifications on algebraic statements (around line 196): It would strengthen the paper to provide brief justification or reference for the following key claims:
> >- “\lambda_i’s must generate the invariant algebra…”
> >- “If the \lambda_i’s and B_i’s are minimal generators, then…”
> >- “Once the model is developed, the coefficient functions C_i’s must be learned….”
>
> The first claim follow directly from the definition of the generators which we mention in Appendix J. The second one follows from Definition (2.2). For the last claim, we have added the appropriate citation [1].
>
> [ 1 ] Robertson, Hovard P. "The invariant theory of isotropic turbulence." Mathematical Proceedings of the Cambridge Philosophical Society. Vol. 36. No. 2. Cambridge University Press, 1940.
>
> >5. On the neural network implementation: Could you discuss how factors such as the number of layers, neurons, initialization, or the scale of training data influence results? Do different architectures affect the learned coefficient functions C_i(\lambda)?
>
> We have added further details about the learning setup in Section 5. Our models employ simple MLPs with two hidden layers and approximately 20k parameters. Increasing the capacity of these networks did not significantly affect performance in our experiments. It would be interesting to explore whether more complex architectures for learning the coefficient functions could yield additional improvements, but this investigation is beyond the scope of the current paper and will be pursued in future work.
>
> >6. On non-uniqueness: Since neural networks typically do not yield unique solutions due to nonconvex optimization, how does this non-uniqueness or local convergence behavior affect the learned tensor function representation?
>
> For the examples we considered, we used the AdamW optimizer with a weight decay of $1e^{-4}$. We did not observe any training instabilities, and the ensembles exhibited low variance in test errors.
>
> >7. Minor issue: In line 417, there is a missing blank space.
>
> We fixed the typo.

---

### Official Review · Reviewer_YY2F · 2025-11-01

**Soundness:** 3
**Presentation:** 3
**Contribution:** 2
**Rating:** 4
**Confidence:** 3

**Summary:**

This paper presents a mathematically grounded framework for learning equivariant tensor function representations under the orthogonal group SO(3). The approach leverages the isomorphism between symmetric trace-free tensors (harmonic tensors) and binary forms to derive minimally complete and equivariant tensor function representations. By combining tools from invariant theory (notably Gordan’s algorithm and covariant algebra) with a learning framework for coefficient functions, the paper constructs models that are interpretable, efficient, and theoretically minimal. The method is validated through the 1) recovery of classical tensor function results for symmetric 2nd-order tensors and 2) a challenging application in turbulence modeling that is learning nonlinear models for the Rapid Pressure Strain Rate (RPSR) tensor, where it outperforms traditional and linear models.

**Strengths:**

**Originality**

The paper introduces a novel synthesis of algebraic invariant theory (via binary forms and covariant modules) and modern learning frameworks to obtain minimal equivariant tensor representations.

This bridges a notable gap between symbolic invariant methods (e.g., Gordan/Hilbert theory) and equivariant neural networks (ENNs), which are data-driven but often opaque and computationally heavy.

**Quality**

The mathematical treatment is sound, detailed, and references foundational results (Hilbert, Olive, Spencer, etc.).

Algorithm 1 (for minimal generator reduction using graded Nakayama’s lemma) is an elegant and efficient contribution addressing a nontrivial computational bottleneck.

Empirical validation on turbulence data is convincing, achieving significant error reductions compared to classical models (LRR and IP) and linear model.

**Clarity**

The exposition of theoretical sections is rigorous and precise, though dense.

**Significance**

Potentially highly relevant to physics-informed machine learning, with potential applications in materials science, continuum mechanics, and turbulence modeling.

**Weaknesses:**

Its originality is primarily mathematical, not algorithmic or conceptual in the machine learning sense (no new architecture, loss, or learning paradigm).

Computations using Gordan’s algorithm and transvectants may explode combinatorially, no empirical evidence that this scales beyond low-order tensors.

The empirical validation is narrow and domain-specific (turbulence modeling only). There are no results across multiple datasets, tasks, or noise regimes, and no ablation studies on the learned coefficients or architecture choices. It is also lacking comparisons to modern equivariant neural networks (E(3)-equivariant GNNs, Tensor Field Networks, etc.)

**Questions:**

Can the authors benchmark their method against modern equivariant deep learning baselines on public datasets? Even a small-scale comparison (accuracy, runtime, interpretability) could help establish the framework’s competitive or complementary nature.

What is the computational complexity of generating covariant modules for larger tensor orders?

Can the framework extend beyond SO(3) to SE(3) or other transformation groups relevant in ML?

How is the “learning” part concretely implemented? How can it be integrated into modern ML pipelines? (architecture design, demonstration of end-to-end differentiability, etc.)

---

> ### Author Response · Authors · 2025-12-03
> **Response to Reviewer YY2F**
>
> ## **Addressing Weaknesses**
>
> >Its originality is primarily mathematical, not algorithmic or conceptual in the machine learning sense (no new architecture, loss, or learning paradigm).
>
> While our work is grounded in mathematical theory, it goes beyond just theory and we believe that it is relevant in the machine learning field: We introduces a systematic and general framework for constructing interpretable and finite equivariant tensor functions with arbitrary index symmetries. Another key contribution is that trace constraints are enforced exactly by means of harmonic decomposition. In contrast, enforcing these constraints in standard neural network architectures would typically require additional penalty terms or specialized losses, none of which are needed in our approach.
>
> As reviewers JCzB and sKrU note, our aim is to position the method as a hybrid design: we first establish the underlying algebraic structure, which guarantees the desired equivariance, symmetries, and trace constraints, and later integrate this structure into a learnable model that can be fine tuned using neural networks and data.
> We believe these are novel methods which are broadly applicable for a number of applications.
>
> >Computations using Gordan’s algorithm and transvectants may explode combinatorially, no empirical evidence that this scales beyond low-order tensors.
>
>
> >The empirical validation is narrow and domain-specific (turbulence modeling only). There are no results across multiple datasets, tasks, or noise regimes, and no ablation studies on the learned coefficients or architecture choices. It is also lacking comparisons to modern equivariant neural networks (E(3)-equivariant GNNs, Tensor Field Networks, etc.)
>
> It is true that for cases involving a large number of higher order input tensors and modeling at higher degrees, our techniques might become intractable. However, we have deployed several strategies to ensure computational feasibility in all stages of our work. We refer you to the guidelines which we recommend guidelines for scalability in a response to Question 2 of reviewer sKrU.
>
> Additionally, the computations required for Gordan’s algorithm and for constructing the covariant module generators only need to be performed once and can then be stored. Once the generators of the invariant ring and covariant module are available, they can be reused for any other problem with the same input–output tensor orders, requiring only the learning of new coefficient functions.
>
> Finally, we note that the existing literature faces significant limitations in constructing equivariant tensor functions, even at relatively low degrees, when tensors of order greater than two are involved. In contrast, such constructions are straightforward using our framework. We consider explicit comparisons with ENNs to be out of scope for the current work.

---

> ### Author Response · Authors · 2025-12-03
> **Responses to Reviewer YY2F (continued)**
>
> ## **Response to Questions**
>
> >Can the authors benchmark their method against modern equivariant deep learning baselines on public datasets? Even a small-scale comparison (accuracy, runtime, interpretability) could help establish the framework’s competitive or complementary nature.
>
> We consider extensive bench-marking with public datasets and modern architectures out of scope for the current work,
> However, we report the results for testing our method against [1]. In [1], a model with 2 million parameters is used to test the O(3) equivariance on a toy problem with increasing amounts of training set sizes.
> Using our method, we first identify the generators of the relevant ring and covariant module. Then we use a simple MLP with 2 hidden layers to learn the coefficient functions. The model has 20k parameters.
> We observe that using our methodology, for training sets of sizes greater than 10,000 samples, our model performs at least as good as [1], while using a much smaller model.
>
> >What is the computational complexity of generating covariant modules for larger tensor orders?
>
> The apriori tenor analysis and computations for the problem presented in the paper took less than 10 mins on a Macbook M1 Max.
> There are several computational bottlenecks which had to overcome to ensure the feasibility of our method. We address several scalability issues our response for the Q2 posed by Reviewer sKrU. Please refer to that for further details.
>
> >Can the framework extend beyond SO(3) to SE(3) or other transformation groups relevant in ML?
>
> We address the generality of our framework in a newly added section. We refer you to Appendix J.
>
> >How is the “learning” part concretely implemented? How can it be integrated into modern ML pipelines? (architecture design, demonstration of end-to-end differentiability, etc.)
>
> The problem of modeling has been split into learnable and fixed parts. The learning part concerns the scalar coefficient functions which are learned using MLPs with tensor invariants as inputs. After the coefficients are calculated the tensor model is constructed by multiplying the coefficients with the tensor monomials and comparing it against the ground truth tensor to get the training loss. Simple MLP architectures are anticipated to be sufficient. We used of Torch to build our models entirely. So, it is anticipated that out complete model can be integrated into an architecture that requires differentiability without any issues.

---

### Author Response · Authors · 2025-12-03
**Summary of Major Revisions and Clarifications**

We thank all reviewers for their thoughtful and constructive feedback. We sincerely appreciate the time and effort you put into your reviews. They have been extremely helpful in improving the clarity and presentation of our work. All revisions have been highlighted in blue in the updated manuscript. Below, we summarize the major changes and address recurring questions.

1. **Clarity**: Several sections of the paper have been rewritten to improve the clarity of the presentation. The introduction and contributions have been refined to state the problem and motivations more directly. The results sections (Sections 4 and 5) have been updated to ensure consistent nomenclature throughout the manuscript. The appendices have also been substantially expanded, with additional mathematical background and illustrative examples added where appropriate.
2. **Generality**: The current work focuses on tensor representations of the SO(3) group. Extensions to O(3) and to permutation group equivariant functions have been added in Appendix I. Moreover, the general ansatz developed in the paper applies to other groups and their representations. This general ansatz is also elaborately discussed in Appendix J.
3. **Comparisons against baselines**: Extensive comparisons with public datasets and existing ENN architectures are beyond the scope of the current work and shall be pursued in future research. However, we evaluated our approach on the toy O(3)-equivariance task from [1] and observed that, given sufficient data (training set size $>$ 1e4), our method achieves comparable performance while using significantly smaller MLPs for learning coefficient functions.

[1] Villar, Soledad, et al. "Scalars are universal: Equivariant machine learning, structured like classical physics." Advances in neural information processing systems 34 (2021): 28848-28863.

---

### Meta-Review · Area_Chair_MtMc · 2026-01-02

**Summary:**

The paper concerns the problem of learning elements in the module of SO(3)-equivariant tensor-valued polynomial functions of tensor arguments (over the associated invariant ring). Exploiting the isomorphism between binary forms and symmetric trace-free (harmonic) tensors, it presents a novel algorithmic approach for computing a (finite) minimal generating sets of this module. With these generators fixed, the learning problem reduces to estimating scalar coefficient functions on a generating set of invariants, which are learned using neural networks. The proposed framework is validated both on classical representations of tensor functions and on a nontrivial application to turbulence modeling.

Reviewers broadly regard the paper as technically sound and mathematically elegant, and they appreciate the use of interpretable representations together with the potential for performance gains. The turbulence application is viewed as a meaningful and nontrivial case study that demonstrates the framework’s promise. At the same time, several recurring concerns limit the paper’s suitability for acceptance at ICLR in its current form. In particular, multiple reviewers found the presentation challenging for a general machine learning audience. While the authors made genuine efforts to improve clarity and expand the appendix, it remains uncertain whether these accessibility issues were fully resolved during the rebuttal. More importantly, reviewers noted that the empirical validation is lacking broad benchmarking against equivariant neural network baselines or evaluations across multiple datasets or tasks. Although such comparisons were requested during the review process, the authors indicated that they fell outside the intended scope of the present work.

Taken together, while the paper is potentially impactful, the reviewer feedback highlights opportunities to further engage with contemporary machine learning practice and evaluation, which could enhance its fit with ICLR.

**Reviewer Concerns:**

Addressed: The rebuttal improved clarity and presentation by revising the introduction, standardizing notation, and expanding the appendix with additional background and examples. The authors also clarified the scope of the contribution, the role of minimal generators, and the learning setup for the scalar coefficient functions, addressing several technical and expository concerns.

Outstanding: The main concern regarding the narrow empirical validation remains. \
Some concerns about accessibility for a general machine learning audience may also persist despite the revisions.

**Reviewer Scores:**

Any projection of score changes is necessarily speculative, but based on the written reviews and the rebuttal, my assessment is as follows. Reviewers who were already positive about the mathematical contribution and primarily raised concerns about clarity (o5pb, sKrU) might modestly increase or maintain their scores given the expanded appendix and clarifications. Reviewers whose scores were below threshold due to the narrow empirical evaluation and lack of benchmarking against equivariant neural network baselines (YY2F, JCzB) are unlikely to raise their scores, as these concerns were not substantively addressed and were explicitly deemed out of scope by the authors. Overall, it appears unlikely that the discussion would have shifted the overall score distribution decisively above the acceptance threshold.

---

### Decision · Program_Chairs · 2026-01-26

Reject